# AlignCLIP: Self-Guided Alignment for Remote Sensing Open-Vocabulary Semantic Segmentation

## Abstract

Open-Vocabulary Semantic Segmentation (OVSS) for remote sensing imagery plays a crucial role in applications such as land cover mapping and environmental monitoring. Recently, Contrastive Language-Image Pre-training (CLIP) has advanced the *training-free* paradigm of OVSS while also inspiring its exploration in the remote sensing domain. However, directly applying CLIP to remote sensing leads to cross-modal mismatches. Prevalent methods focus on exploring attention mechanism of CLIP visual encoder or introducing vision foundation models to obtain more discriminative feature, but they often overlook the alignment between patches and textual representations. To address this issue, we propose a *training-free* framework named **AlignCLIP**. We find that, objects of the same category tend to exhibit a more compact distribution in remote sensing, this enables a single visual feature to effectively represent all objects within the category. Based on this observation, we design the *Self-Guided Alignment (SGA)* module, which leverages the most reliable text-specific visual prototypes to refine the text embeddings. To mitigate interference among irrelevant features, we further introduce the *Cluster-Constrained Enhancement (CCE)* module, which clusters semantically similar patch features, suppresses inter-cluster correlations, and updates the logits map via a constraint propagation mechanism. Experiments on eight remote sensing benchmarks demonstrate that AlignCLIP consistently outperforms state-of-the-art *training-free* OVSS methods, achieving an average gain of +2.2 mIoU and offering a robust adaptive solution for open-vocabulary semantic segmentation in remote sensing. All code will be released.

## 1 Introduction

Open-vocabulary semantic segmentation (OVSS) in remote sensing imagery serves as a fundamental task in land cover mapping and environmental monitoring. Using arbitrary textual descriptions, it enables pixel-level classification of remote sensing images. The remarkable success of Contrastive Language–Image Pre-training (CLIP) (Radford et al., 2021b) in zero-shot recognition has inspired the development of OVSS. Most prior studies have focused on fine-tuning CLIP (Liang et al., 2023; Peng et al.; Wei et al., 2023; Peng et al., 2025; Zeng et al., 2024; Lin et al., 2024; Zhang et al., 2025), but their progress is limited by the demand for large annotated datasets. Moreover, remote sensing imagery often contains categories beyond the training set due to seasonal changes, land use evolution, and geographic diversity, making these approaches difficult to generalize. Recently, several works (Wang et al., 2023a; Yang et al., 2024; Zhou et al., 2022; Lan et al., 2024c) have begun to explore *training-free* paradigms in natural image domain, which achieve OVSS by extracting image patches and textual representations and directly performing cross-modal matching. This paradigm has further inspired its exploration in the remote sensing domain.

Prevalent *training-free* approaches in natural image domain primarily focus on the image modality, and they explore the attention mechanism of the CLIP visual encoder or integrate advanced vision foundation models (VFMs) to obtain more discriminative features (Lan et al., 2024b; Shao et al., 2024; Kim et al., 2025b; Barsellotti et al., 2024). However, these methods largely overlook the alignment between image patches and textual representations. Most existing open-vocabulary segmentation methods perform mask-category recognition by aligning region-level features with

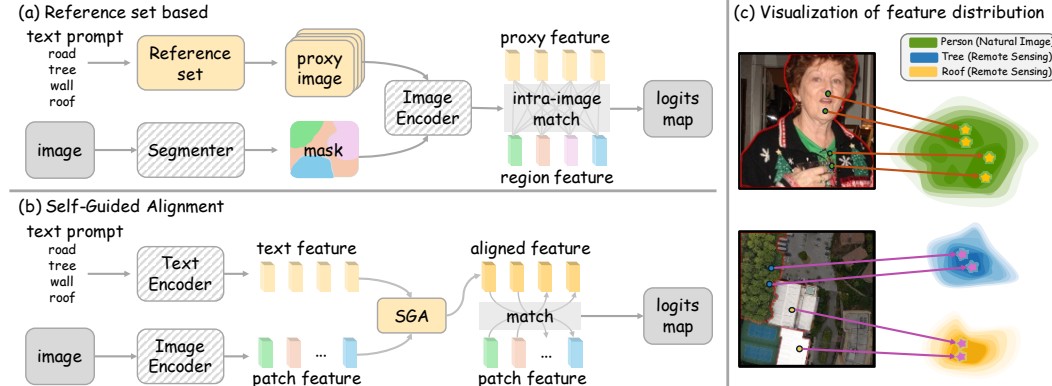

Figure 1: (a) **Reference-set** paradigm primarily focuses on constructing an accurate image–text matching set and performing matching based on proxy images. (b) **Our SGA module** refines the textual representation by selecting the most reliable visual prototypes from patch features and encourages mismatched patches to align with their corresponding textual representations. (c) We present **feature visualization** for both natural and remote sensing images and observe that, compared with natural images, objects of the same category in remote sensing imagery exhibit a more compact distribution.

CLIP-based textual embeddings. Although CLIP demonstrates remarkable generalization capabilities on downstream classification tasks, it often suffers from spatial ambiguity and co-occurrence-induced object confusion, which stem from its holistic pre-training objective and lead to cross-modal mismatches (Wang et al., 2024). To address this issue, another line of research leverages external image–text reference sets (Xuan et al., 2025; Wang et al., 2024), specifically, these approaches transform text–image matching into intra-image matching by retrieving proxy images associated with category texts, as shown in Fig. 1(a), thereby mitigating cross-modal discrepancies. However, they heavily depend on the construction of cumbersome reference sets and exhibit limited generalization to unseen scenarios.

In this work, we observe that compared with natural images, the feature distribution within the same category in remote sensing images is more compact, as illustrated in Fig. 1(c). Intuitively, remote sensing images are captured from a much farther distance than natural images, which dilutes the fine details of objects and thus results in more uniform features. On the other hand, the

Table 1: Statistics of intra-class feature similarity in natural images and remote sensing.

| Image domain | #Pairs | Similarity |
|---|---|---|
| Natural Image | 331,998 | 0.67±0.10 |
| Remote Sensing | 342,799 | 0.89±0.05 |

fixed top-down viewing angle of remote sensing images also contributes to the high similarity of intra-class features (*e.g.*, all water appear blue, and all buildings are represented by rooftops). To further validate this observation, we calculated the intra-class feature similarity of objects in two image domains (*i.e.*, natural image and remote sensing) and report the mean and standard deviation of the similarity, as presented in Table 1. We can find that the mean intra-class feature similarity of remote sensing images is significantly higher than that of natural images, with a smaller standard deviation. This indicates a more compact feature distribution, thereby verifying the rationality of our observation.

Based on this observation, we naturally conceive a solution: selecting the features most similar to the given text features from the image feature space as text-specific visual prototypes, and aligning the text features with the visual prototypes. Due to the high compactness of intra-class visual features in the remote sensing, this alignment enables text features to match their corresponding visual features more stably and accurately. Building upon this, we proposed a simple yet effective *training-free* framework, termed **AlignCLIP** to mitigate cross-modal mismatches in OVSS of remote sensing imagery. We designed two key modules: (a) *Self-Guided Alignment (SGA)*, which leverages the most reliable text-specific visual prototypes of the target image to refine textual embeddings, thereby bringing mismatched patches closer to their correct textual semantics. (b) *Cluster-*

*Constrained Enhancement (CCE)*, which clusters semantically similar patches while suppressing inter-cluster correlations, and updating logits map through constrained propagation.

Notably, AlignCLIP operates in a fully *training-free* manner, thereby eliminating the need for labor-intensive reference sets construction. By relying solely on information inherent to the target image, it further ensures strong generalization across diverse scenarios. Extensive evaluations on eight remote sensing benchmarks demonstrate that AlignCLIP consistently outperforms state-of-the-art *training-free* OVSS methods, highlighting its robustness and adaptability to novel scenarios and unseen categories.

The main contributions of our work are as follows:

- We analyze the limitations of existing reference sets-based methods, and observe that objects of the same category in remote sensing imagery exhibit concentrated feature distribution. Leveraging this characteristic, we mitigate cross-modal mismatches while obviating the need for cumbersome reference sets construction.

- We propose AlignCLIP, a fully *training-free* framework that alleviates cross-modal mismatches. The framework incorporates the *Self-Guided Alignment (SGA)* module, which refines text embeddings using reliable text-specific prototypes, and the *Cluster-Constrained Enhancement (CCE)* module, which clusters image patches and suppresses the correlations between different clusters.

- Extensive experiments on eight remote sensing benchmarks demonstrate that AlignCLIP consistently outperforms state-of-the-art *training-free* OVSS methods, achieving both qualitative and quantitative improvements and exhibiting strong generalization to diverse scenarios and unseen categories.

## 2 RELATED WORK

**Vision-Language Models.** Vision-Language Models (VLMs) (Jia et al., 2021; Yuan et al., 2021) aim to align visual and textual representations within a shared semantic space, enabling zero-shot and open-vocabulary recognition. A landmark advancement in this field is CLIP (Radford et al., 2021b), a dual-encoder trained contrastively on image–text pairs with strong downstream generalization. However, CLIP is optimized for image-level classification, and its patch features are suboptimal for dense prediction (Cheng et al., 2022; Xu et al., 2022) due to limited spatial awareness and the absence of explicit spatial modeling. This issue is more prominent in the remote sensing domain, where high-resolution scenes exhibit fine spectral-textural details and large-scale layouts distinctly different from those of natural images (Cao et al., 2024; Zhang et al., 2025; Dutta et al., 2025; Fu et al., 2025; 2024). Although some works (*e.g.*, RemoteCLIP (Liu et al., 2024), GeoRSCLIP (Zhang et al., 2024b)) have been adapted to remote sensing via prompt engineering or fine-tuning, such approaches typically require task-specific retraining or substantial labeled data, constraining their practicality for open-vocabulary semantic segmentation.

**Vision Foundation Models.** Vision Foundation Models (VFMs) (Caron et al., 2021; Oquab et al., 2023; Siméoni et al., 2025; Kirillov et al., 2023; Ravi et al., 2024) provide general visual representations across a wide range of tasks. One category of such models is DINO (Caron et al., 2021), which learns semantically rich and spatially coherent features via self-distillation. It can localize objects without explicit supervision, making it highly suitable for dense prediction tasks. Additionally, SAM (Kirillov et al., 2023) demonstrates strong image segmentation capabilities, supporting various segmentation prompts (*e.g.*, points, boxes, and masks) with excellent cross-domain generalization performance. In this work, we leverage the representations from VFMs to cluster semantically similar features and utilize inter-cluster correlations to update the logits map.

**Training-free OVSS.** *Training-free* open-vocabulary semantic segmentation (OVSS) labels pixels for arbitrary categories at inference by matching visual and textual embeddings from VLMs like CLIP via cross-modal similarities. prevalent works improve spatial awareness via attention modification (Yang et al., 2024; Wang et al., 2023a) or by integrating VFMs such as SAM (Zhang et al., 2024a; Lan et al., 2024c), but they overlook the correlations between patches and text representations. ReMe (Xuan et al., 2025) mitigate mismatches using curated reference sets, which are costly and difficult to generalize. In the remote sensing domain, SegEarth-OV (Li et al., 2025) represents the first *training-free* OVSS framework, which introduces an upsampling module to adapt CLIP, but

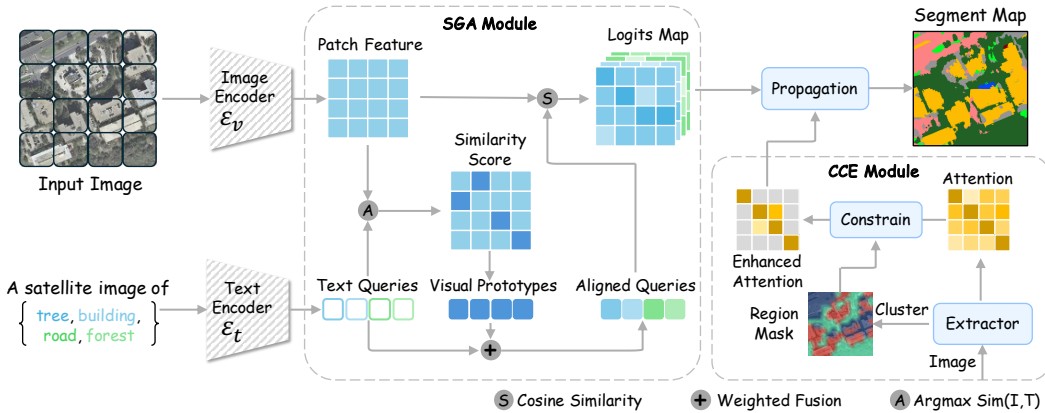

Figure 2: **The overall framework of AlignCLIP.** We propose a simple yet effective approach to alleviate cross-modal mismatches. We design two core modules (a) *Self-Guided Alignment (SGA)*, which refines textual embeddings using the most reliable text-specific visual prototypes. (b) *Cluster-Constrained Enhancement (CCE)*, which clusters semantically similar patches while suppressing inter-cluster correlations and updates the logits map through constrained propagation.

still encounter cross-modal mismatches. Our work inherits the upsampling module of SegEarth-OV and designs two modules to alleviate the cross-modal mismatches, leveraging the characteristic of concentrated intra-class feature distribution in remote sensing imagery.

## 3 METHODOLOGY

In this section, we first introduce a preliminary of our framework in Sec. 3.1. Then, we introduce the *Self-Guided Alignment (SGA)* module in Sec. 3.2 and the *Cluster-Constrained Enhancement (CCE)* module in Sec. 3.3. Finally, we detail the integration with the upsampling module in Sec. 3.4. The overall framework is shown in Fig. 2.

### 3.1 PRELIMINARY

Given a remote sensing input image $I \in \mathbb{R}^{H \times W \times 3}$ and an open set of textual category names $\mathcal{T} = \{t_1, t_2, \ldots, t_K\}$, where $H, W$ denote the height and width of an image, $K$ denotes the number of classes. The objective of open-vocabulary semantic segmentation (OVSS) is to assign each pixel in $I$ to one of the categories in $\mathcal{T}$.

In the *training-free* setting, recent works adopt large-scale vision-language models (*e.g.*, ViT-based CLIP) as the backbone for feature extraction and cross-modal matching. Specifically, the frozen CLIP image encoder $\mathcal{E}_v$ divides the input image $I$ into a grid of image patches and outputs a set of patch-level visual embeddings. For brevity, we omit the $[CLS]$ token here:

$$\mathbf{P} = [\mathbf{p}_1, \mathbf{p}_2, \ldots, \mathbf{p}_N] \in \mathbb{R}^{N \times D}, \tag{1}$$

where $D$ denotes the embedding dimension and $N = H_p \times W_p$ depends on the encoder's patch resolution. However, the CLIP model has limited capability in spatial awareness, previous studies have modified the attention score calculation in the last layer of self-attention in the CLIP visual encoder from *query-to-key* to *query-to-query* or *key-to-key*, which has significantly improved the performance of CLIP's dense prediction. Following the practice of prior works (Li et al., 2025), we modified the calculation of the self-attention scores in the last layer of the visual encoder:

$$MSA(\boldsymbol{q}, \boldsymbol{k}, \boldsymbol{v}) = \sum_{i \in \{\boldsymbol{q}, \boldsymbol{k}, \boldsymbol{v}\}} softmax(\frac{i \cdot i^T}{\sqrt{d}}) \cdot \boldsymbol{v}, \tag{2}$$

where $\boldsymbol{q}$, $\boldsymbol{k}$ and $\boldsymbol{v}$ represent the *query*, *key*, and *value* matrices in self-attention, respectively, and $d$ denotes the dimension of attention features. Meanwhile, to obtain more accurate text embeddings,

we adopt a prompt template that is more suitable for remote sensing scenarios (*e.g.*, "a satellite image of [CLS].") to incorporate contextual information, as opposed to the prompt template used for natural images. Subsequently, each text prompt is processed by the CLIP text encoder $\mathcal{E}_t$ to obtain its corresponding textual embedding:

$$\mathbf{T} = [\mathbf{t}_1, \mathbf{t}_2, \ldots, \mathbf{t}_K] \in \mathbb{R}^{K \times D}. \tag{3}$$

Finally, we compute the logits map between each patch-level visual embedding $p_i \in \mathbf{P}$ and all textual embeddings $\mathbf{T}$ using cosine similarity. The segmentation mask is obtained by applying the **argmax** operation to logits map:

$$\mathcal{S} = sim(\mathbf{P}, \mathbf{T}), \quad \mathcal{S} \in \mathbb{R}^{N \times K}. \tag{4}$$

## 3.2 Self-Guided Alignment

In the process of cross-modal matching, the inherent gap between text and image leads to cross-modal mismatches. To address this, we design a *Self-Guided Alignment (SGA)* module, which exploits the intrinsic visual cues from the reliable text-specific visual prototypes to refine the textual embeddings. Formally, for each textual embedding $\mathbf{t}_k \in \mathbf{T}$, we compute its cosine similarity with all patch-level visual embeddings $\mathbf{p}_i \in \mathbf{P}$:

$$s_{i,k} = \frac{\mathbf{p}_i^\top \mathbf{t}_k}{\|\mathbf{p}_i\| \|\mathbf{t}_k\|}. \tag{5}$$

We then select the most similar patch embedding $\mathbf{p}_{i^*}$ for category $k$:

$$i^* = \arg\max_i \ s_{i,k}. \tag{6}$$

The selected patch embedding serves as a text-specific visual prototype directly extracted from the target image. Owing to the high intra-class compactness observed in remote sensing imagery, this visual prototype naturally clusters with other features of the same category in the feature space. By aligning the text embedding $\mathbf{t}_k$ with this prototype $\mathbf{p}_{i^*}$, we effectively reduce their feature space discrepancy, enabling the aligned text embedding $\mathbf{t}_k'$ to match its corresponding visual features more stably and accurately:

$$\mathbf{t}_k' = (1 - \alpha) \cdot \mathbf{t}_k + \alpha \cdot \mathbf{p}_{i^*}, \tag{7}$$

where $\alpha \in [0, 1]$ is a balancing hyperparameter controlling the contribution of textual and visual components.

Finally, the logits map for segmentation is computed by replacing the original textual embeddings with the aligned embeddings $\mathbf{T}' = \{\mathbf{t}_1', \ldots, \mathbf{t}_K'\}$:

$$\mathcal{S}' = sim(\mathbf{P}, \mathbf{T}'). \tag{8}$$

Notably, since the prototypes are derived on-the-fly from the target image, the SGA module naturally adapts to new scenes without requiring any re-training or prebuilt reference sets.

## 3.3 Cluster-Constrained Enhancement

Although the SGA module mitigates cross-modal mismatches by refining text embeddings with text-specific visual prototypes, the image patches may still be disturbed by irrelevant patches. To address this issue, we introduce the *Cluster-Constrained Enhancement (CCE)* module, which aggregates semantically similar patches, suppresses interactions between irrelevant patches, and updates logits map through constrained propagation.

Specifically, we employ a VFM visual transformer (*e.g.*, DINO, SAM) to extract a high discriminative visual feature map $\mathbf{F} \in \mathbb{R}^{H_p \times W_p \times D}$ from the target image. We reshape $\mathbf{F}$ into $N$ patch embeddings $\{\mathbf{f}_1, \ldots, \mathbf{f}_N\}$ and apply a clustering algorithm:

$$\{\mathcal{C}_m\}_{m=1}^{K_c} = \text{Clustering}(\{\mathbf{f}_i\}_{i=1}^N), \tag{9}$$

where $\mathcal{C}_m$ denotes the set of patch indices assigned to cluster $m$, and $K_c$ is the total number of clusters–a hyperparameter controlling the granularity of segmentation refinement.

In addition to visual features, we also extract the self-attention matrix $\mathbf{A} \in \mathbb{R}^{N \times N}$ from the final layer of the Vision Transformer, which encodes pairwise affinities between patches. However, directly applying this matrix to propagate information over the logits map can be detrimental, as the affinities between different semantic categories are generally non-zero, thereby introducing undesired cross-category interactions. To address this issue, we employ clustering results to mask the attention matrix, we retain affinities between patches belonging to similar categories while setting the affinities between dissimilar categories to zero. Specifically, we construct a binary clustering mask matrix $\mathbf{M} \in \{0,1\}^{N \times N}$ as follows:

$$\mathbf{M}_{ij} = \begin{cases} 1, & \text{if } \mathcal{G}(i) = \mathcal{G}(j), \\ 0, & \text{otherwise,} \end{cases} \tag{10}$$

where $\mathcal{G}(i)$ denotes the cluster assignment of patch $i$. The masked affinity matrix $\tilde{\mathbf{A}}$ is then refined as:

$$\tilde{\mathbf{A}} = \mathbf{A} \odot \mathbf{M}, \tag{11}$$

where $\odot$ denoting element-wise multiplication. In this way, affinities are preserved only within the same cluster, while inter-cluster correlations are suppressed, logits map are propagated under the cluster-constrained affinities as follows:

$$\hat{\mathcal{S}} = \text{Norm}(\tilde{\mathbf{A}} \cdot \mathcal{S}'). \tag{12}$$

During the propagation process, the logits maps are weighted and averaged based on affinity values within the same cluster. This ensures that logits maps from different clusters do not interfere with each other, while logits maps within the same cluster maintain consistent semantic predictions, ultimately resulting in more accurate mask predictions.

### 3.4 INTEGRATION WITH UPSAMPLING MODULE

To recover the fine-grained details critical for accurate segmentation in high-resolution remote sensing images, we inherit the upsample module from SegEarth-OV. Specifically, the visual feature map $\mathbf{P}$ is first reshaped into a 2D feature representation $\mathbf{P} \in \mathbb{R}^{H_p \times W_p \times D}$, which is subsequently upsampled to the original image resolution. The upsampled features are then computed with the text embeddings $\mathbf{T}$ via cosine similarity, yielding an upsampled logits map:

$$\mathcal{S}_{up} = sim(featup(\mathbf{P}), \mathbf{T}). \tag{13}$$

And then, we interpolate the $\hat{S}$ to match the spatial size of $\mathcal{S}_{up}$, the two logits maps are then fused via a weighted combination:

$$\mathcal{S}_{final} = \beta \cdot Interpolate(\hat{\mathcal{S}}) + (1 - \beta) \cdot \mathcal{S}_{up}, \tag{14}$$

where $\beta \in [0, 1]$ is a fusion weight controlling the balance between CCE-refined logits map and upsampled logits map. $Interpolate$ is a bilinear interpolation algorithm.

Finally, we apply an **argmax** operation over $\mathcal{S}_{final}$ to produce the final segmentation mask:

$$pred = \arg \max_k \mathcal{S}_{final}. \tag{15}$$

## 4 EXPERIMENTS

### 4.1 SETTINGS

**Datasets and Evaluation Metric.** We conducted comprehensive experiments on eight widely used remote sensing semantic segmentation datasets. Among these, OpenEarthMap (Wang et al., 2023b), LoveDA (Wang et al., 2021), iSAID (Waqas Zamir et al., 2019), Potsdam (Gerke, 2014) and Vaihingen (Rottensteiner et al., 2014) are primarily composed of satellite images, while UAVid (Yang et al., 2020), UDD5 (Chen et al., 2018) and VDD (Pan et al., 2021) mainly consist of UAV images. These datasets collectively cover diverse spatial resolutions, imaging conditions, and scene types, thereby providing a comprehensive evaluation of model robustness. Each dataset contains multiple foreground categories along with a background category. Please refer to appendix A.1 for detailed

Table 2: Quantitative comparison results on eight remote sensing datasets. **Bold** fonts indicate the optimal results, and underlined fonts indicate the suboptimal results. Avg. represents the average mIoU across the eight datasets.

| Methods | OpenEarthMap | LoveDA | iSAID | Potsdam | Vaihingen | UAVid | UDD5 | VDD | Avg. |
|---|---|---|---|---|---|---|---|---|---|
| CLIP[ICML'21] | 12.0 | 12.4 | 7.5 | 14.5 | 10.3 | 10.9 | 9.5 | 14.2 | 11.4 |
| MaskCLIP[ECCV'22] | 25.1 | 27.8 | 14.5 | 31.7 | 24.7 | 28.6 | 32.4 | 32.9 | 27.2 |
| SCLIP[ECCV'24] | 29.3 | 30.4 | 16.1 | 36.6 | 28.4 | 31.4 | 38.7 | 37.9 | 31.1 |
| GEM[CVPR'24] | 33.9 | 31.6 | 17.7 | 36.5 | 24.7 | 33.4 | 41.2 | 39.5 | 32.3 |
| ClearCLIP[ECCV'24] | 31.0 | 32.4 | 18.2 | 40.9 | 27.3 | 36.2 | 41.8 | 39.3 | 33.4 |
| NACLIP[WACV'25] | 35.7 | 31.5 | 19.5 | 40.2 | 28.8 | 37.5 | 42.1 | 40.9 | 34.5 |
| ResCLIP[CVPR'25] | 34.2 | 31.2 | 20.0 | 42.6 | 28.2 | 37.6 | 42.3 | 40.3 | 34.6 |
| ProxyCLIP[ECCV'24] | 35.0 | 33.5 | 20.7 | 44.1 | 27.8 | 42.1 | 46.5 | 44.3 | 36.8 |
| CASS[CVPR'25] | 34.6 | 34.0 | 20.6 | 42.9 | 31.5 | 38.6 | 39.0 | 40.9 | 35.3 |
| SC-CLIP[ArXiv'24] | 35.9 | 31.7 | 18.4 | 43.4 | 29.6 | 38.3 | 42.0 | 41.0 | 35.0 |
| Trident[ICCV'25] | 35.1 | 31.5 | 20.0 | 44.4 | 27.7 | 41.8 | 44.1 | 45.7 | 36.3 |
| CorrCLIP[ICCV'25] | 35.4 | 32.7 | 16.9 | 42.6 | 24.7 | 38.1 | 40.1 | 37.7 | 33.5 |
| SegEarth-OV[CVPR'25] | 39.8 | 36.9 | 21.7 | 47.1 | 29.1 | 42.5 | 50.6 | 45.3 | 39.1 |
| AlignCLIP-D (Ours) | **40.1** (+0.3) | **39.5** (+2.6) | **23.6** (+1.9) | **47.9** (+0.8) | 34.5 (+3.0) | **44.4** (+1.9) | **51.8** (+1.2) | **48.4** (+2.8) | **41.3** (+2.2) |
| AlignCLIP-S (Ours) | 40.1 (+0.3) | 39.5 (+2.6) | 23.4 (+1.7) | 47.8 (+0.7) | **34.6** (+3.1) | 44.4 (+1.9) | 51.8 (+1.2) | 48.1 (+2.8) | 41.2 (+2.1) |

dataset information. Following common practice in semantic segmentation, we report the mean Intersection over Union (mIoU) as the primary evaluation metric, which provides a balanced measure of classification accuracy across categories.

**Baselines.** We compared our AlignCLIP with a wide range of state-of-the-art *training-free* OVSS methods, including CLIP (Radford et al., 2021b), MaskCLIP (Zhou et al., 2022), SCLIP (Wang et al., 2023a), GEM (Bousselham et al., 2024), ClearCLIP (Lan et al., 2024a), NACLIP (Hajimiri et al., 2024), ResCLIP (Yang et al., 2024), ProxyCLIP (Lan et al., 2024c), CASS (Kim et al., 2025a), SC-CLIP (Bai et al., 2025), Trident (Shi et al., 2024) and CorrCLIP (Zhang et al., 2024a). These baselines represent different design paradigms such as attention modification and proxy-based adaptation. Furthermore, we evaluated SegEarth-OV, a method specifically tailored for remote sensing that employs a trained upsampling module to recover the lost detailed information in feature maps. It should be noted that the performance of reference-set-based methods (*e.g.*, ReME) largely depends on the scale and quality of the constructed reference set, making fair comparisons challenging. Therefore, we do not report evaluations of these methods in the experimental section.

**Implementation Details.** We provide two model variants of AlignCLIP, *i.e.*, AlignCLIP-D (integration with DINO) and AlignCLIP-S (integration with SAM). All experiments employ Open-CLIP (Radford et al., 2021a) to extract both image and text features. Unless otherwise specified, all models adopt ViT-B/16 as the default backbone. For the text encoder, we adopted a remote-sensing-oriented prompt template, with the prompt list provided in appendix A.2. For the image encoder, we followed the settings of SegEarth-OV, input images were resized such that the long side was 448, and inference was conducted using a sliding window of size $224 \times 224$ with a stride of 112. For the clustering algorithm, we simply used the K-Means algorithm (Ikotun et al., 2023) with the number of clusters $K_c = 3$ as default. For the specific balance ratios $\alpha$ and fusion weights $\beta$ of each dataset, please refer to appendix A.3. To isolate the effectiveness of our method, all post-processing techniques (*e.g.*, PAMR (Araslanov & Roth, 2020), denseCRF (Krähenbühl & Koltun, 2011)) were disabled. Experiments were conducted on 8 RTX 3090 GPUs, and all the code of our implementation is based on **mmsegmentation** repository[1].

## 4.2 RESULTS

**Quantitative Evaluation.** As shown in Table 2, AlignCLIP achieves the overall best performance across all eight remote sensing benchmarks, achieving a highest average mIoU of **41.3%**, which outperforms all compared *training-free* OVSS methods. The improvements are particularly remark-

---

[1]https://github.com/open-mmlab/mmsegmentation

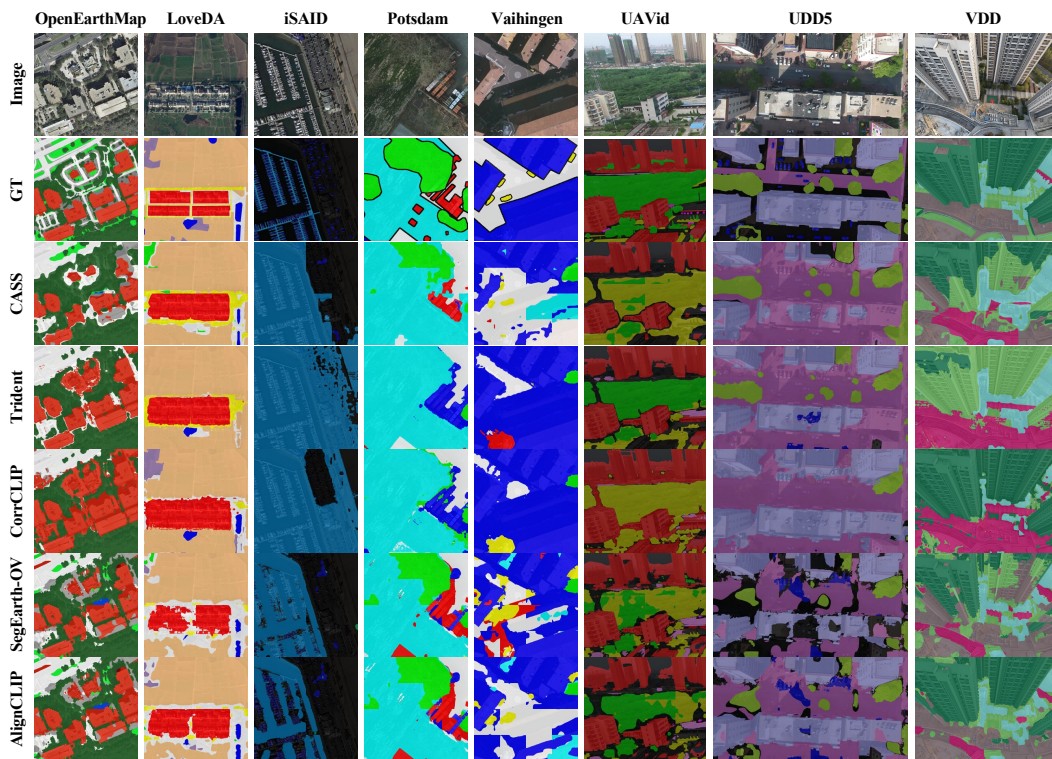

Figure 3: Qualitative comparison of different training-free OVSS methods on eight remote sensing datasets.

able on datasets such as LoveDA (**+2.6%**), Vaihingen (**+3.1%**), and VDD (**+2.8%**). On the remaining datasets, including OpenEarthMap, iSAID, Potsdam, UAVid, and UDD5, our method also achieves steady gains over existing approaches. Moreover, compared to SegEarth-OV, AlignCLIP still achieves a substantial improvement (**+2.2%** on average). Furthermore, we observe that the performance improvement varies significantly across different datasets. For instance, the Open-EarthMap only achieves a 0.3% improvement compared to the baseline method. We attribute this to the large scale of images in the OpenEarthMap dataset, which results in relatively small proportions of certain categories (*e.g.*, buildings) within the images. During the feature extraction process, features of small objects are easily overlooked, which impairs the selection of visual prototype and leads to a degradation in the logits map, resulting in limited performance gain. Interestingly, the two model variants based on DINO and SAM yield comparable results, which demonstrates that our method exhibits robust performance across the two mainstream VFM architectures. The above experimental results demonstrate that our method achieves consistent improvements across different scenarios and model architectures.

**Qualitative Evaluation.** As illustrated in Fig. 3, we present the qualitative visualization results of AlignCLIP-D and other representative methods. The results demonstrate that compared with CASS, Trident, CorrCLIP, and SegEarth-OV, our AlignCLIP generates more accurate and spatially coherent segmentation results across various datasets. Existing methods often suffer from category confusion (*e.g.*, walls vs. roofs in VDD). While SegEarth-OV improves boundary smoothness, it still exhibits matching errors in fine-grained structures. In contrast, AlignCLIP effectively mitigates cross-modal mismatches, producing clearer regions and sharper object boundaries. For more qualitative comparisons, refer to Appendix A.11.

### 4.3 ABLATION STUDIES

In this section, we conduct a comprehensive ablation study to evaluate the effectiveness of the proposed components in AlignCLIP and to examine the impact of key hyperparameters. For clarity, the

hyperparameter sensitivity analysis reports results on four representative datasets—OpenEarthMap (OEM), Potsdam (PD), UAVid (UAV) and VDD, while the complete experimental results are provided in Appendix A.4. Unless otherwise specified, we use the AlignCLIP-D as the default model for our primary analysis.

**Component ablation analysis.** We first investigate the effectiveness of our proposed SGA and CCE modules in AlignCLIP through component-wise ablation, as reported in Table 3. The $1^{st}$ row reports the performance of the baseline method SegEarth-OV, the $2^{nd}$ and $3^{rd}$ rows report the results of introducing the SGA and CCE modules respectively, while the $4^{th}$ row reports the performance of the complete model. We can find that: **i)** incorporating the SGA module alone yields a 1.0% improvement, demonstrating that alleviating cross-modal mismatches can substantially enhance segmentation performance. **ii)** applying the CCE module alone provides only a modest 0.3% gain, we attribute this to the fact that the CCE module operates on the aligned logits map produced by the SGA module, and propagating optimization on a poorly aligned logits map offers limited benefit.

Table 3: Ablation analysis of different components.

| SGA | CCE | OpenEarthMap | LoveDA | iSAID | Potsdam | Vaihingen | UAVid | UDD5 | VDD | Avg. |
|---|---|---|---|---|---|---|---|---|---|---|
| | – | 39.8 | 36.9 | 21.7 | 47.1 | 29.1 | 42.5 | 50.6 | 45.3 | 39.1 |
| ✓ | | 39.8 | 37.8 | 22.7 | 47.2 | 32.3 | 43.3 | 50.6 | 46.8 | 40.1 ↑1.0 |
| | ✓ | 39.8 | 34.9 | 21.1 | 47.3 | 31.6 | 43.2 | 51.2 | 45.8 | 39.4 ↑0.3 |
| ✓ | ✓ | **40.1** | **39.5** | **23.6** | **47.9** | **34.5** | **44.4** | **51.8** | **48.4** | **41.3** ↑2.2 |

**Effect of the balance ratios $\alpha$.** We further study the effect of the balance ratios $\alpha$, which control the relative contributions of visual and textual features, a larger $\alpha$ assigns greater weight to the visual features. As shown in Table 4a, we observe that increasing $\alpha$ does not lead to a monotonic performance gain, instead, the performance generally rises initially and then declines (*e.g.*, when $\alpha = 0.3$ on the PD dataset). We attribute this phenomenon to the fact that, as the contribution of patch features increases, the text features can better align with the image features. However, beyond a certain threshold—determined by the feature distribution of the dataset, the image features begin to compromise the general representational capacity of the text features.

**Effect of the fusion weights $\beta$.** We further investigate the effect of the fusion weights $\beta$, which control the balance between our logits map and the upsampling logits map, a larger $\beta$ indicates a smaller contribution of the upsampled logits map. As shown in Table 4b, smaller values (*e.g.*, $\beta = 0.1$) achieve the best performance on the OEM and PD datasets, while larger values (*e.g.*, $\beta = 0.3$ and $\beta = 0.4$) yield better results on the UAV and VDD datasets. We attribute this result to the differences in dataset scales, the OEM and PD datasets consist of large-scale satellite images, where detail information is more likely to be lost during feature extraction, thus requiring more contributions from the upsampled logits map to compensate. In contrast, UAV and VDD datasets contain small-scale UAV aerial images, where detail information is relatively preserved, making the contribution of the upsampled logits map relatively limited.

Table 4: Sensitivity analysis of various hyperparameters across different datasets.

(a) Balance ratios $\alpha$

| $\alpha$ | OEM | PD | UAV | VDD |
|---|---|---|---|---|
| 0.1 | **40.1** | 47.5 | 44.0 | 47.7 |
| 0.2 | **40.1** | 47.6 | **44.4** | 48.2 |
| 0.3 | 39.8 | **47.9** | 44.3 | 48.3 |
| 0.4 | 39.3 | 47.8 | 43.9 | 47.9 |
| 0.5 | 38.8 | 47.7 | 43.3 | 47.2 |

(b) Fusion weights $\beta$

| $\beta$ | OEM | PD | UAV | VDD |
|---|---|---|---|---|
| 0.1 | **40.1** | **47.9** | 43.4 | 46.7 |
| 0.2 | 40.0 | 47.8 | 44.0 | 47.9 |
| 0.3 | 39.6 | 47.5 | **44.4** | **48.4** |
| 0.4 | 39.1 | 46.9 | **44.4** | **48.4** |
| 0.5 | 38.6 | 46.0 | 44.3 | 48.2 |

(c) Cluster numbers $K_c$

| $K_c$ | OEM | PD | UAV | VDD |
|---|---|---|---|---|
| 3 | **40.1** | **47.9** | **44.4** | **48.4** |
| 6 | 40.1 | 47.6 | 44.2 | 48.2 |
| 9 | 40.0 | 47.6 | 44.2 | 48.1 |
| 12 | 40.0 | 47.6 | 44.1 | 48.0 |
| 15 | 40.0 | 47.5 | 44.0 | 47.9 |

**Effect of the cluster number $K_c$.** We further analyze the effect of varying number of the clusters $K_c$ used in the CCE module (*i.e.*, $K_c = 3, 6, 9, 12, 15$), with the results summarized in Table 4c. The results indicate that $K_c = 3$ achieves the best performance across all datasets. Increasing the number of clusters leads to a slight performance decrease, with a drop of no more than 0.5%, suggesting that our method is not sensitive to the choice of $K_c$. We attribute this to the fact that $K_c = 3$ is sufficient

for the model to distinguish irrelevant features, and further increasing the number of clusters does not yield significant performance gains.

**Effect of different top-$n$ visual prototypes.** We further investigate the effect of varying the number of visual prototypes on text embeddings. Specifically, we compute the cosine distance between each patch feature and the text feature, select the $n$ closest patch features (*i.e.*, $n = 1, 2, 3, 4, 5, 6$), and average them before fusing with the text features. As illustrated in Fig. 4, our experiments reveal that increasing the number of patch features leads to a consistent performance decline across all eight datasets. This indicates that excessive prototypes not only fail to improve patch–text alignment but also introduce mismatched features, thereby increasing noise. Therefore, we select only the most similar feature to pursue the best performance.

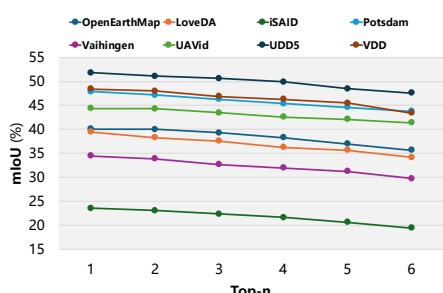

Figure 4: The effect of different numbers of top-$n$ visual prototypes.

## 5 CONCLUSION

In this work, we presented AlignCLIP, a novel *training-free* framework for open-vocabulary semantic segmentation in the remote sensing domain. We find that features of intra-class objects in remote sensing tend to be compact. Based on this observation, we design two modules to alleviate cross-modal mismatches between image patches and textual representations. Specifically, the *Self-Guided Alignment (SGA)* module leverages the most reliable text-specific visual prototypes to refine textual embeddings, and the *Cluster-Constrained Enhancement (CCE)* clusters semantically similar patches while suppressing inter-cluster correlations, and updating logits map through constrained propagation. Extensive experiments across eight remote sensing benchmarks demonstrated that AlignCLIP consistently outperforms state-of-the-art approaches, We hope this work can inspire future related research and bring new possibilities to *training-free* open-vocabulary semantic segmentation in the remote sensing domain.

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

# A APPENDIX

## A.1 DATASET DESCRIPTION

**OpenEarthMap** contains 5,000 aerial and satellite remote sensing images, including 8 foreground classes and 1 background class, with a spatial resolution of 0.25-0.5 meters, covering 97 regions in 44 countries / territories on six continents. we use the validation set for evaluation.

**LoveDA** includes remote sensing images with a spatial resolution of 0.3 meters covering multiple cities, totaling 5,987 images. These images are annotated with 6 foreground classes and one background class. we use the validation set for evaluation.

**iSAID** consists of images captured by the JL-1 satellite and GF-2 satellite. It includes 15 foreground classes and one background class. Following the data processing pipeline of MMSegmentation, we cropped the images into rectangles with a size of 896 and an overlapping area of 384. Finally, 33,978 images were generated for training and 11,644 for validation. In this study, the validation set was used for evaluation.

**Potsdam** comprises 38 image patches with a spatial resolution of 0.05 meters, with an average size of 6,000×6,000 pixels. It includes 5 foreground categories and 1 background category. Following the data processing pipeline of MMSegmentation, we used the validation set for evaluation.

**Vaihingen** comprises 33 image patches with a spatial resolution of 0.09 meters, with an average size of 2,494×2,064 pixels. It includes 5 foreground categories and 1 background category. Following the data processing pipeline of MMSegmentation, we used the validation set for evaluation.

**UAVid** is a 4K semantic segmentation video dataset for urban scenes, which contains a large number of street views and is annotated with 6 foreground classes and 1 background class. In this study, its test set was used for evaluation.

**UDD5** consists of images collected by unmanned aerial vehicles (UAVs), including 4 foreground classes and 1 background class. In this study, we used its validation set for evaluation.

**VDD** is a collection of UAV images featuring diverse scenes, camera angles, and varying weather/lighting conditions. It provides high-resolution annotated images at the 400-pixel scale. With 6 foreground classes and 1 background class, its test set was used for evaluation in this study.

## A.2 REMOTE SENSING PROMPT TEMPLATE

To obtain more effective text embeddings for remote sensing scenarios, we carefully designed 80 prompt templates tailored to remote sensing scenarios to replace the prompt templates oriented to natural images. As shown in Table 5, five representative examples are presented. For each category, these prompt templates are used to generate corresponding text features, which are then averaged to obtain a semantically rich category feature representation for semantic segmentation.

Table 5: Examples of remote sensing prompt templates for generating text descriptions.

| Remote sensing prompt templates |
| :---: |
| a low-quality aerial image of [class]. |
| a cropped remote sensing image of [class]. |
| a remote sensing interpretation map of [class]. |
| a satellite image containing hardly recognizable [class]. |
| a low-resolution remote sensing image of [class]. |

## A.3 HYPERPARAMETER SETTING

We provide the detailed hyperparameter settings for each dataset corresponding to the two model variants, *i.e.*, AlignCLIP-D and AlignCLIP-S, as shown in Table 6. The balance ratio $\alpha$ denotes

the contribution of text features and visual features, while the fusion weight $\beta$ is used to control the fusion weight between the CCE-refined logits and the upsampled logits.

Table 6: Hyperparameter settings of AlignCLIP across different datasets.

| hyperparameters | OpenEarthMap | LoveDA | iSAID | Potsdam | Vaihingen | UAVid | UDD5 | VDD |
|---|---|---|---|---|---|---|---|---|
| **Integration with DINO** | | | | | | | | |
| $\alpha$ | 0.1 | 0.7 | 0.5 | 0.3 | 0.3 | 0.2 | 0.3 | 0.2 |
| $\beta$ | 0.1 | 0.2 | 0.2 | 0.1 | 0.5 | 0.3 | 0.2 | 0.3 |
| **Integration with SAM** | | | | | | | | |
| $\alpha$ | 0.1 | 0.5 | 0.5 | 0.3 | 0.3 | 0.2 | 0.3 | 0.2 |
| $\beta$ | 0.1 | 0.2 | 0.2 | 0.2 | 0.5 | 0.4 | 0.2 | 0.4 |

## A.4 SENSITIVITY ANALYSIS DETAILS

In this section, we present the detailed sensitivity analysis of the hyperparameters involved in the two model variants (*i.e.*, AlignCLIP-D and AlignCLIP-S) across eight datasets, as shown in Table 7-9. Specifically, the balance ratios $\alpha$ is used to control the contribution of text features and visual features in the SGA module (see Sec. 3.2), the fusion weights $\beta$ is employed to control the fusion balance between the CCE-refined logits map and the upsampled logits map (see Sec. 3.4), and $K_c$ represents the number of clusters in the clustering algorithm within the CCE module, which is used to control the granularity of segmentation refinement (see Sec. 3.3).

Table 7: Sensitivity analysis of different balance ratios $\alpha$ across different datasets.

| $\alpha$ | OpenEarthMap | LoveDA | iSAID | Potsdam | Vaihingen | UAVid | UDD5 | VDD | Avg. |
|---|---|---|---|---|---|---|---|---|---|
| **Integration with DINO** | | | | | | | | | |
| 0.1 | **40.1** | 36.2 | 21.7 | 47.5 | 32.7 | 44.0 | 51.5 | 47.7 | 40.2 |
| 0.2 | **40.1** | 37.3 | 22.3 | 47.6 | 33.9 | **44.4** | 51.7 | **48.4** | 40.7 |
| 0.3 | 39.8 | 38.2 | 22.9 | **47.9** | **34.5** | 44.3 | **51.8** | 48.3 | **40.9** |
| 0.4 | 39.3 | 38.8 | 23.4 | 47.8 | 34.4 | 43.9 | **51.8** | 47.9 | **40.9** |
| 0.5 | 38.8 | **39.2** | **23.6** | 47.7 | 34.0 | 43.3 | 51.7 | 47.2 | 40.7 |
| **Integration with SAM** | | | | | | | | | |
| 0.1 | **40.1** | 36.1 | 21.6 | 47.5 | 32.7 | 44.0 | 51.4 | 47.4 | 40.1 |
| 0.2 | 40.0 | 37.2 | 22.2 | 47.7 | 33.9 | **44.4** | 51.7 | **48.1** | 40.7 |
| 0.3 | 39.9 | 38.1 | 22.7 | **47.8** | **34.6** | **44.4** | **51.8** | **48.1** | **40.9** |
| 0.4 | 39.3 | 38.8 | 23.2 | **47.8** | 34.5 | 43.9 | **51.8** | 47.7 | **40.9** |
| 0.5 | 38.7 | **39.2** | 23.4 | **47.8** | 34.1 | 43.4 | 51.7 | 47.0 | 40.7 |

Table 8: Sensitivity analysis of different fusion weights $\beta$ across different datasets.

| $\beta$ | OpenEarthMap | LoveDA | iSAID | Potsdam | Vaihingen | UAVid | UDD5 | VDD | Avg. |
|---|---|---|---|---|---|---|---|---|---|
| **Integration with DINO** | | | | | | | | | |
| 0.1 | **40.1** | 38.5 | 22.2 | **47.9** | 31.1 | 43.4 | 51.4 | 46.7 | 40.1 |
| 0.2 | 40.0 | **39.5** | **23.6** | 47.8 | 32.5 | 44.0 | **51.8** | 47.9 | 40.9 |
| 0.3 | 39.6 | 39.3 | **23.6** | 47.5 | 33.4 | **44.4** | **51.8** | **48.4** | **41.0** |
| 0.4 | 39.1 | 38.4 | 22.2 | 46.9 | 34.1 | **44.4** | 51.4 | **48.4** | 40.6 |
| 0.5 | 38.6 | 37.4 | 20.4 | 46.0 | **34.5** | 44.3 | 50.9 | 48.2 | 40.0 |
| **Integration with SAM** | | | | | | | | | |
| 0.1 | **40.1** | 38.4 | 22.1 | 47.8 | 31.1 | 43.4 | 51.4 | 46.6 | 40.1 |
| 0.2 | 40.0 | **39.5** | 23.4 | **48.0** | 32.5 | 44.0 | **51.8** | 47.7 | 40.9 |
| 0.3 | 39.6 | 39.3 | 23.4 | 47.8 | 33.4 | 44.4 | **51.8** | 48.1 | **41.0** |
| 0.4 | 39.1 | 38.5 | 22.1 | 47.2 | 34.1 | **44.5** | 51.4 | **48.2** | 40.6 |
| 0.5 | 38.6 | 37.4 | 20.4 | 46.3 | **34.6** | 44.4 | 50.9 | 47.9 | 40.1 |

Table 9: Sensitivity analysis of different cluster numbers $K_c$ across different datasets.

| $K_c$ | OpenEarthMap | LoveDA | iSAID | Potsdam | Vaihingen | UAVid | UDD5 | VDD | Avg. |
|---|---|---|---|---|---|---|---|---|---|
| | **Integration with DINO** | | | | | | | | |
| 3 | **40.1** | **39.5** | **23.6** | **47.9** | **34.5** | **44.4** | **51.8** | **48.4** | **41.3** |
| 6 | **40.1** | 39.3 | 23.5 | 47.6 | 34.4 | 44.2 | **51.8** | 48.2 | 41.1 |
| 9 | 40.0 | 39.2 | 23.5 | 47.6 | 34.3 | 44.2 | 51.7 | 48.1 | 41.1 |
| 12 | 40.0 | 39.1 | 23.4 | 47.6 | 34.1 | 44.1 | 51.6 | 48.0 | 41.0 |
| 15 | 40.0 | 39.0 | 23.3 | 47.5 | 34.1 | 44.0 | 51.6 | 47.9 | 41.0 |
| | **Integration with SAM** | | | | | | | | |
| 3 | **40.1** | **39.5** | **23.4** | 47.8 | **34.6** | **44.4** | **51.8** | **48.1** | **41.2** |
| 6 | **40.1** | 39.3 | **23.4** | 47.7 | 34.4 | 44.3 | 51.7 | **48.1** | 41.1 |
| 9 | 40.0 | 39.2 | 23.3 | 47.6 | 34.3 | 44.2 | 51.6 | 48.0 | 41.0 |
| 12 | 40.0 | 39.1 | 23.3 | 47.6 | 34.2 | 44.2 | 51.6 | 47.9 | 41.0 |
| 15 | 40.0 | 39.0 | 23.3 | 47.6 | 34.1 | 44.1 | 51.5 | 47.8 | 40.9 |

## A.5 SEAMLESS INTEGRATION INTO OTHER METHODS

In this section, we further validate the generality of the proposed approach by integrating the SGA module into other representative frameworks and conducting comprehensive evaluations on eight remote sensing benchmark datasets. Specifically, we incorporate SGA as an independent, plug-and-play component into existing methods. Since SGA aligns only the text embeddings without altering the remaining architecture, it can be seamlessly integrated into a variety of CLIP-based frameworks. For the experimental setup, we select ProxyCLIP, SC-CLIP, and CorrCLIP as baseline methods, with the hyperparameter $\alpha$ uniformly set to 0.1.

As shown in Table 10. The experimental results demonstrate that, across different baseline models, incorporating our method enables the text embeddings to achieve more precise alignment with the patch features. This finding not only confirms that enhancing image–text alignment can significantly improve semantic segmentation performance, but also highlights the generality of the SGA module, which can be seamlessly integrated into other CLIP-based frameworks.

Table 10: The proposed SGA module is integrated as a plugin into other methods.

| Methods | OpenEarthMap | LoveDA | iSAID | Potsdam | Vaihingen | UAVid | UDD5 | VDD | Avg. |
|---|---|---|---|---|---|---|---|---|---|
| ProxyCLIP | 35.0 | 33.5 | 20.7 | 44.1 | 27.8 | 42.1 | 46.5 | 44.3 | 36.8 |
| +SGA | **38.6** | **34.2** | **21.6** | **44.6** | **32.7** | **42.4** | **48.3** | **45.3** | **38.5** ↑1.7 |
| SC-CLIP | 35.9 | 31.7 | 18.4 | 43.4 | 29.6 | 38.3 | 42.0 | 41.0 | 35.0 |
| +SGA | **39.8** | **32.8** | **19.6** | **43.6** | **31.6** | **39.6** | **46.0** | **42.3** | **36.9** ↑1.9 |
| CorrCLIP | 35.4 | 32.7 | 16.9 | 42.6 | 24.7 | 38.1 | 40.1 | 37.7 | 33.5 |
| +SGA | **36.6** | **33.5** | **18.6** | **43.8** | **27.9** | **39.9** | **41.1** | **39.6** | **35.1** ↑1.6 |

## A.6 COMPUTATIONAL ANALYSIS

In this section, we conduct a computational analysis to validate the efficiency and practicality of our proposed method. Specifically, we report the average inference time per image and memory consumption of the baseline method and our two model variants across eight datasets using 8 RTX 3090 GPUs, as presented in Table 11. It can be observed that when solely adopting the SGA module, our method incurs nearly negligible overhead in terms of inference time and memory consumption compared to SegEarth-OV, while achieving a 1% performance improvement. When solely using the CCE module, the inference time increases by an insignificant few milliseconds, although the memory consumption exhibits a relatively more noticeable increase, the combined use of the CCE module with the SGA module yields a 2.2% performance gain, which we consider a favorable trade-off.

Table 11: Computational analysis of AlignCLIP.

| Methods | Time(ms/image) ↓ | Memory(MB) ↓ | Performance(mIoU) ↑ |
|---|---|---|---|
| Trident | 89 | 2514 | 36.3 |
| CorrCLIP | 97 | 2890 | 33.5 |
| SegEarth-OV | 12 | 1392 | 39.1 |
| **Integration with DINO** | | | |
| +SGA | 12 | 1392 | 40.1 |
| +CCE | 16 | 2661 | 39.4 |
| Ours | 16 | 2661 | 41.3 |
| **Integration with SAM** | | | |
| +SGA | 12 | 1392 | 40.1 |
| +CCE | 18 | 2782 | 39.3 |
| Ours | 18 | 2782 | 41.2 |

## A.7 RESULTS ON NATURAL IMAGES

In this section, we perform cross-domain validation on natural images. Specifically, we select five representative natural image segmentation datasets (*i.e.*, Cityscapes, ADE20k, COCO-Stuff, Context59, and VOC20), and integrate our SGA module into three state-of-the-art (SOTA) methods for natural images. The experimental results are presented in Table 12. It can be observed that all three methods exhibit consistent performance degradation, which is consistent with our expectations. These results demonstrate that natural images with scattered intra-class features cannot alleviate cross-modal mismatch by searching for representative visual prototypes. In contrast, our method achieves SOTA performance in remote sensing scenarios with compact intra-class features, further validate the rationality of our motivation.

Table 12: Quantitative comparison results on natural images.

| Methods | Cityscapes | ADE20k | Stuff | Context59 | VOC20 | Avg. |
|---|---|---|---|---|---|---|
| SC-CLIP | 41.0 | 20.1 | 26.6 | 40.1 | 84.3 | 42.4 |
| +SGA | 38.5 | 20.0 | 26.4 | 40.0 | 77.9 | 40.6 |
| Trident | 42.9 | 21.9 | 28.3 | 42.2 | 84.5 | 44.0 |
| +SGA | 38.9 | 21.1 | 27.7 | 42.1 | 82.0 | 42.4 |
| CorrCLIP | 49.9 | 26.9 | 31.6 | 48.8 | 88.8 | 49.2 |
| +SGA | 47.8 | 26.1 | 31.1 | 47.2 | 85.8 | 47.6 |

## A.8 COMBINED WITH POST-PROCESSING

In this section, we present the quantitative results of our method when combined with a post-processing technique. In semantic segmentation, post-processing typically refines the predicted masks by leveraging low-level cues (*e.g.*, color consistency and spatial proximity) through iterative optimization, and it generally leads to performance improvements. In our experiments, we apply denseCRF to the logits maps produced by our method. As shown in Table 13, all datasets exhibit consistent performance gains, resulting in an overall improvement of 0.9% in the average mIoU.

Table 13: Quantitative comparison results based on post-processing.

| Methods | OpenEarthMap | LoveDA | iSAID | Potsdam | Vaihingen | UAVid | UDD5 | VDD | Avg. |
|---|---|---|---|---|---|---|---|---|---|
| Ours | 40.1 | 39.5 | 23.6 | 47.9 | 34.5 | 44.4 | 51.8 | 48.4 | 41.3 |
| +denseCRF | **40.9** | **40.2** | **24.3** | **48.7** | **35.1** | **45.3** | **52.9** | **49.9** | **42.2** ↑0.9 |

## A.9 VISUALIZATION ANALYSIS OF DIFFERENT IMAGE DOMAINS

In this section, we visualize the visual features of objects from several remote sensing and natural image datasets. As shown in Fig. 5, we extract image features using CLIP-B/16 and project them into a two-dimensional space using t-SNE algorithm. The results reveal that, in the natural-image domain, features from different categories tend to overlap substantially, leading to ambiguous class boundaries. In contrast, in the remote-sensing domain, features belonging to the same class form notably more compact clusters and exhibit much less confusion with other categories. This observation provides additional evidence supporting the validity and motivation of our approach.

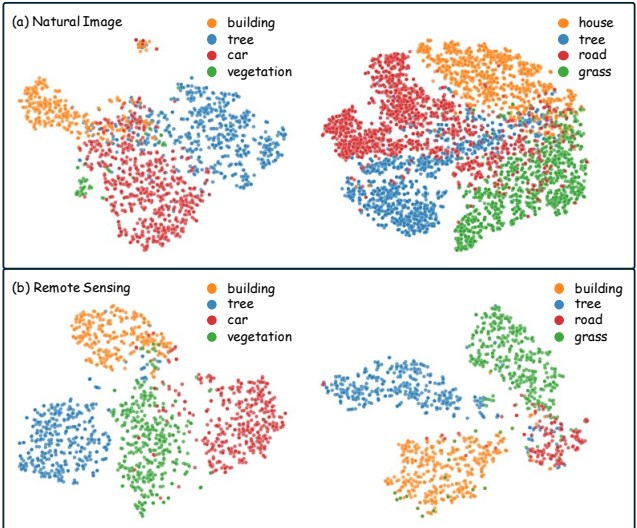

Figure 5: Visualization of intra-class features in different image domains.

## A.10 PSEUDO CODE OF OUR ALIGNCLIP

To clearly present the implementation details of our method and ensure reproducibility, we provide pseudo code for the two core modules of AlignCLIP, *i.e.*, SGA and CCE, in Algorithm 1 and Algorithm 2, respectively. In addition, the full implementation of our method (based on PyTorch), is provided in the supplementary materials, and the complete code will be publicly released after curation.

**Algorithm 1** Pseudo code for Self-Guided Alignment in a PyTorch-like style.

```python
def self_guided_alignment(image_features, query_features, visual_query_alpha):
    """
    Self-Guided Alignment (SGA) module.

    Args:
        image_features: [num_patches, feature_dim]
        query_features: [num_queries, feature_dim]
        visual_query_alpha: balance ratio (0~1)

    Returns:
        Aligned query features: [num_queries, feature_dim]
    """
    num_queries = len(query_features)

    # Similarity between image patches and query features
    similarity = (image_features @ query_features.T).permute(1,0).softmax(dim=-1)
    _, index = similarity.topk(1, dim=-1)

    # Gather top patch features and average
    visual_query_features = torch.gather(
        image_features.unsqueeze(0).repeat(num_queries, 1, 1),
        dim=1,
        index=index.unsqueeze(-1).repeat(1, 1, image_features.shape[-1])
    ).mean(dim=1)

    # Fuse with visual features
    aligned_query_features = visual_query_alpha * visual_query_features + \
                        (1 - visual_query_lambda) * query_features
    return aligned_query_features / aligned_query_features.norm(dim=-1, keepdim=True)
```

**Algorithm 2** Pseudo code for Cluster-Constrained Enhancement in a PyTorch-like style.

```python
def cluster_constrained_enhancement(vfm_features, logits_map, cluster_num):
    """
    Cluster-Constrained Enhancement (CCE) Module.

    Args:
        vfm_features: Feature map for clustering, shape [num_patches, feature_dim].
        logits_map: original logits map, shape [num_patches, num_classes].
        cluster_num: Number of clusters to group patches.

    Returns:
        refined logits map: [num_patches, num_classes].
    """

    # Cluster the features
    _, cluster_ids = perform_clustering(vfm_features, n_clusters=cluster_num)

    # Calculate patch attention
    vfm_attn = vfm_features @ vfm_features.T

    % Calculate masked attn based on clustering results
    masked_attn = torch.zeros_like(vfm_attn)

    for cluster_id in np.unique(cluster_ids):
        # Create mask for current cluster
        mask = (cluster_ids == cluster_id)

        # Aggregate attention within cluster
        masked_attn[mask] = vfm_attn[mask, :] * mask[None, :]   # element-wise masking

    #Propagate attention to refine logits map
    refined_logits = propagate_aff(logits_map, aff=final_attn)

    return refined_logits
```

## A.11  ADDITIONAL QUALITATIVE RESULTS

We provide additional visualization analysis results for eight datasets to further validate the effectiveness of our proposed method, as illustrated in Fig. 6-13.

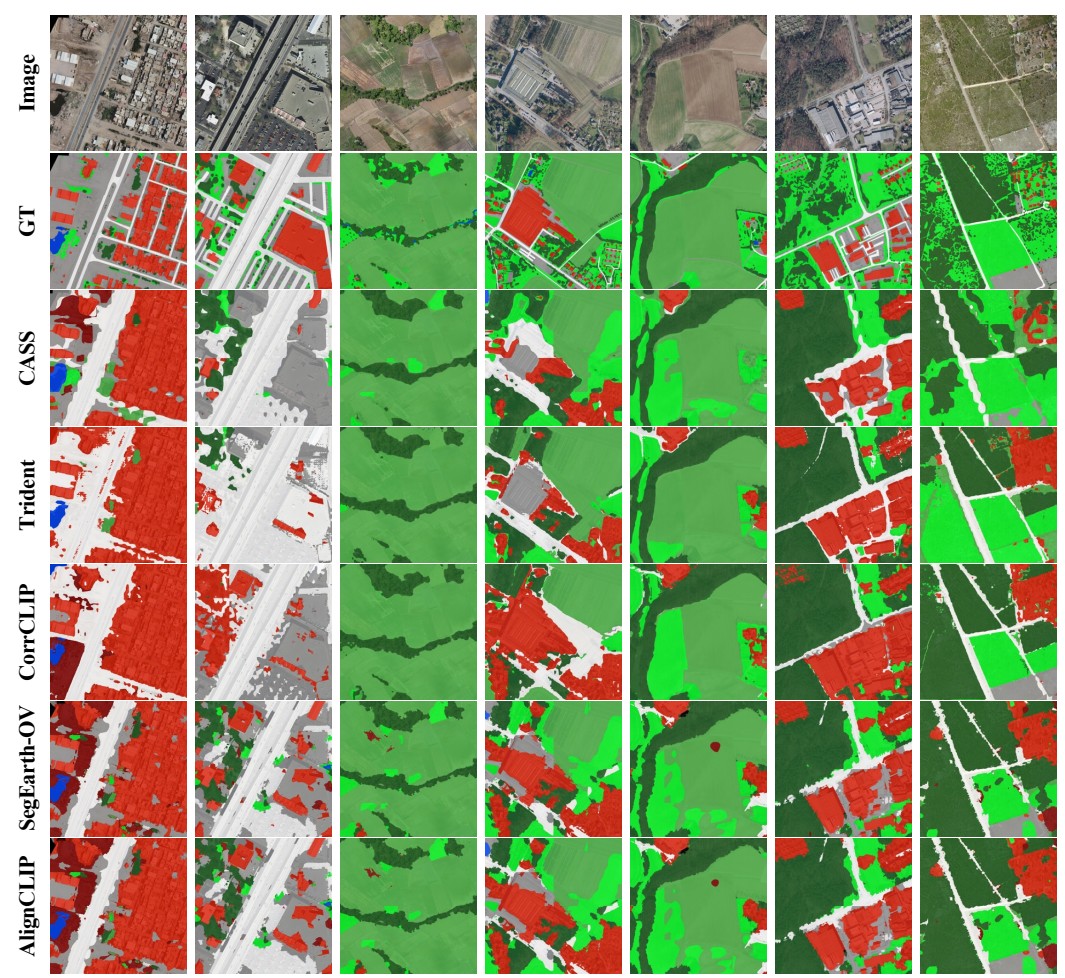

Figure 6: Qualitative comparison of different training-free OVSS methods on OpenEarthMap.

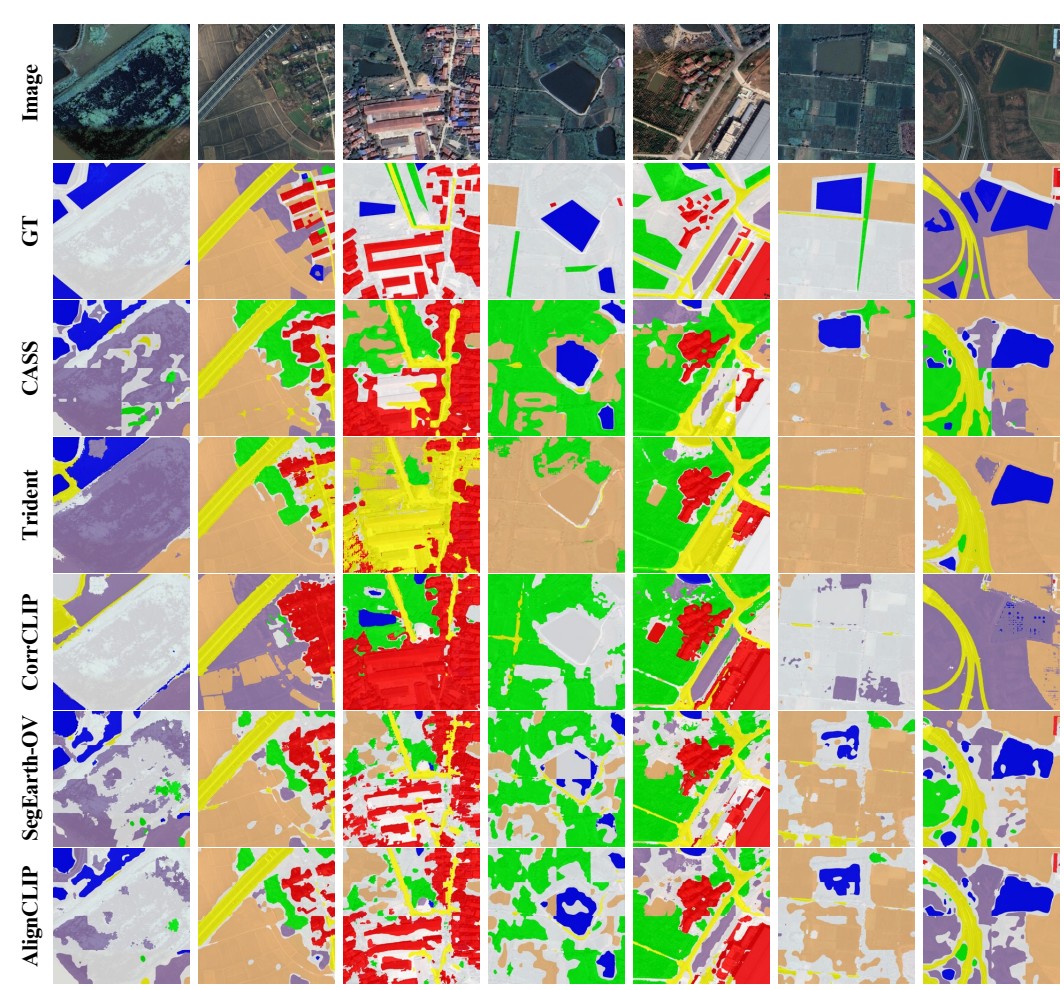

Figure 7: Qualitative comparison of different training-free OVSS methods on LoveDA.

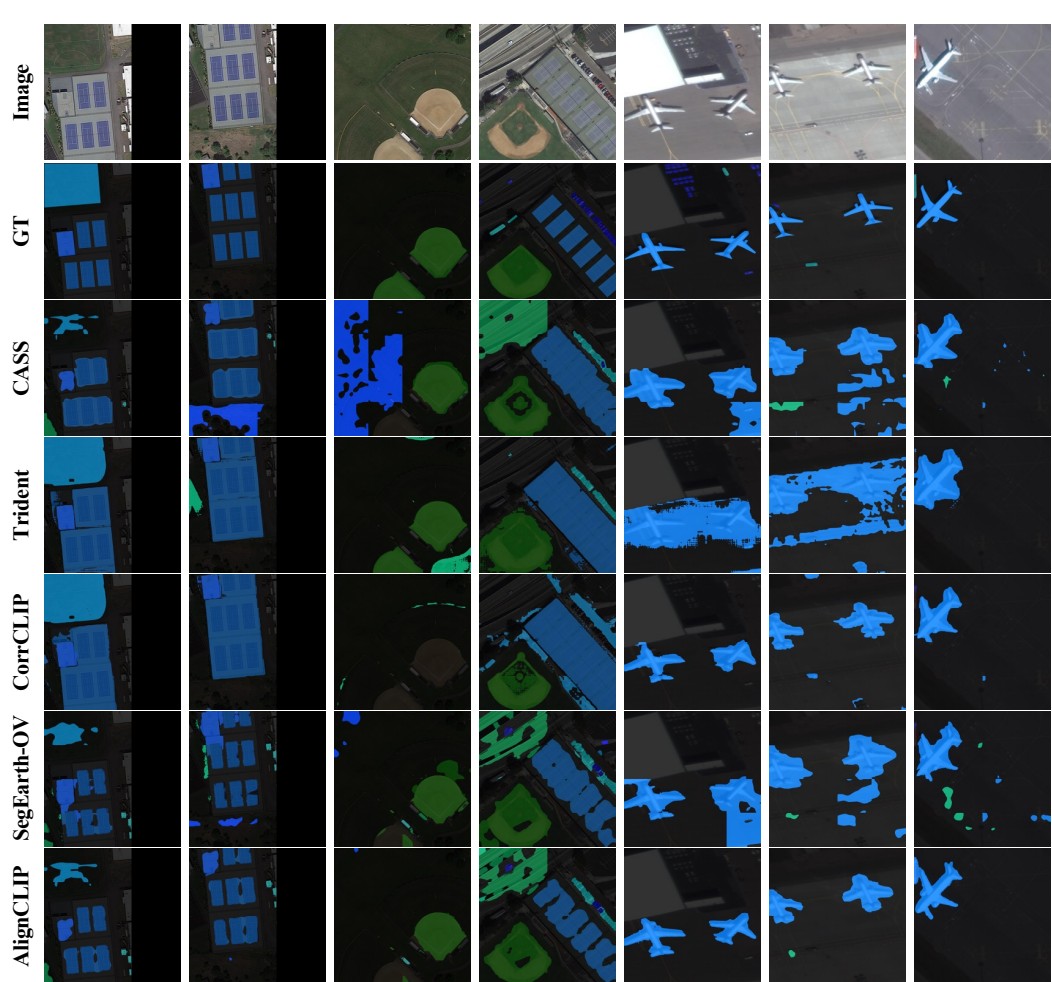

Figure 8: Qualitative comparison of different training-free OVSS methods on iSAID.

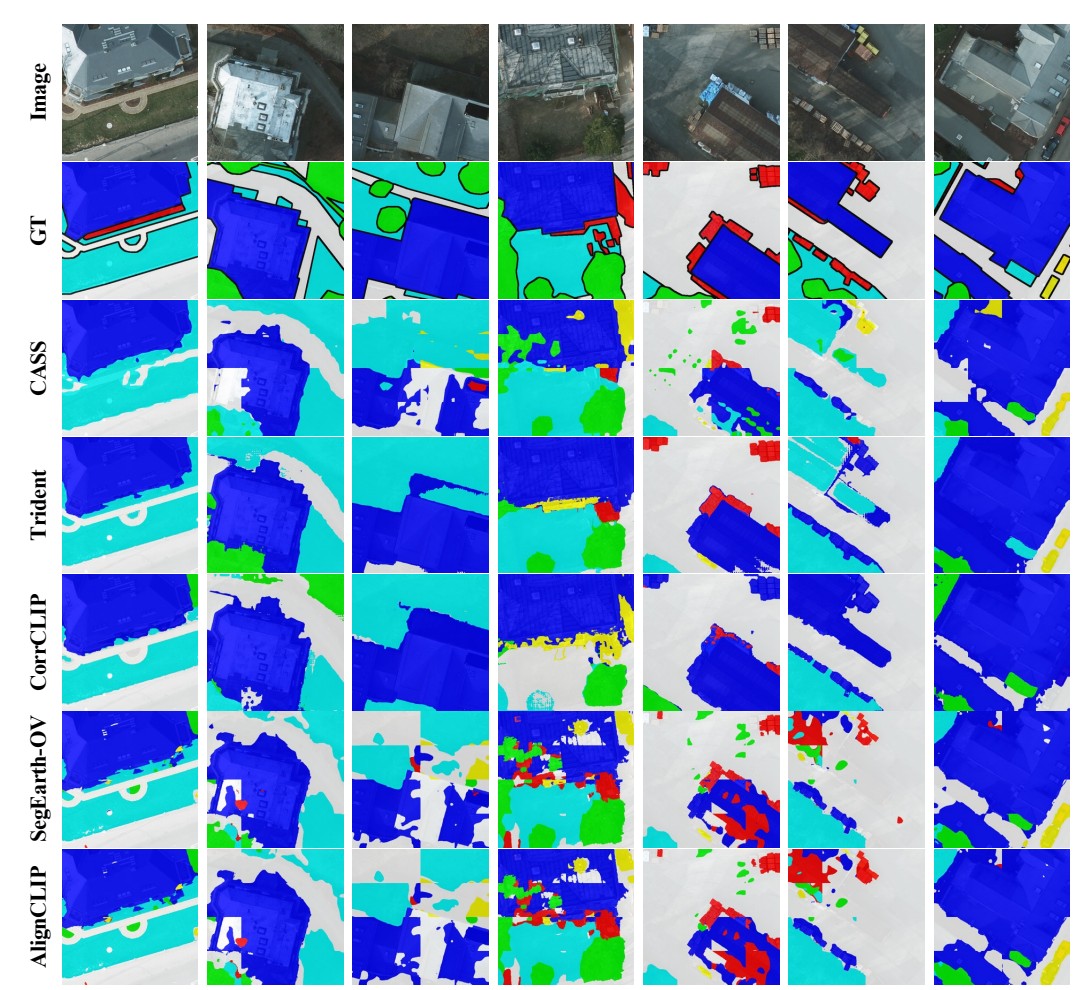

Figure 9: Qualitative comparison of different training-free OVSS methods on Potsdam.

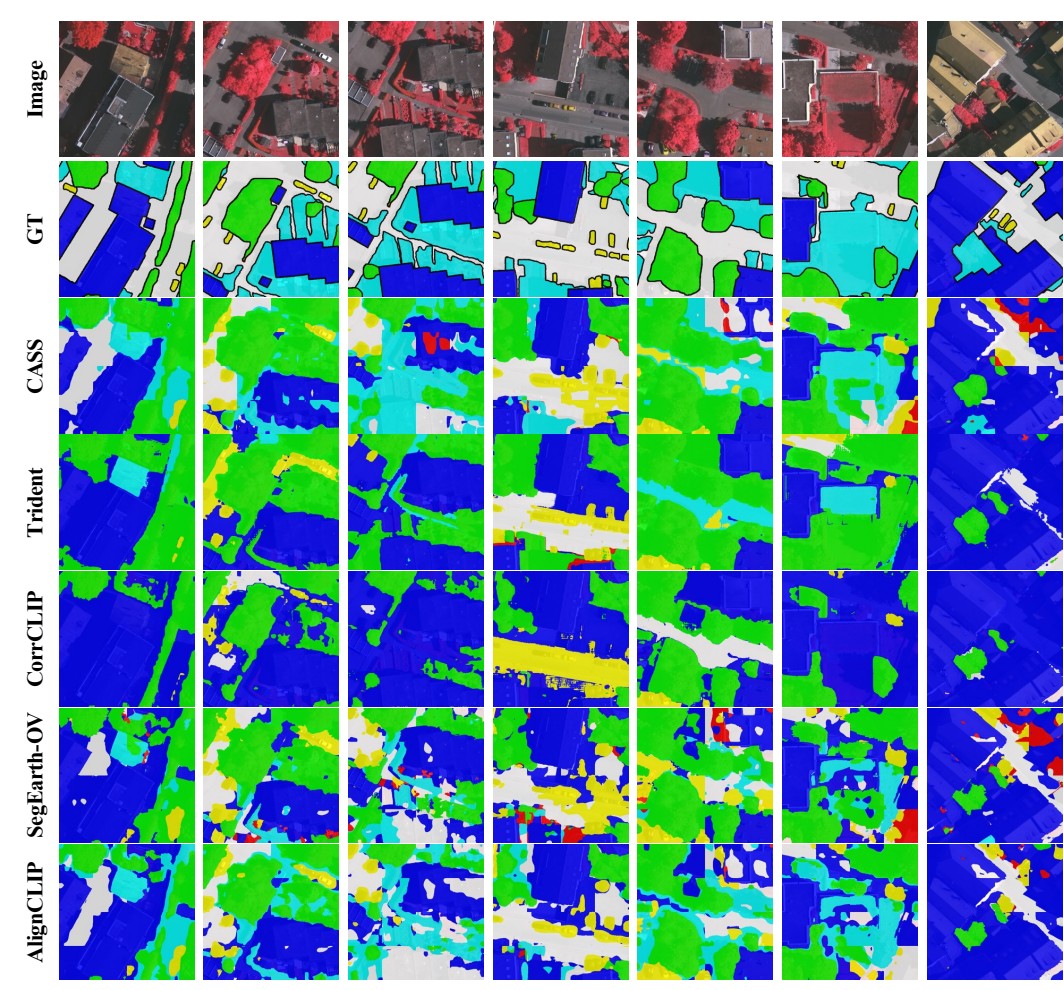

Figure 10: Qualitative comparison of different training-free OVSS methods on Vaihingen.

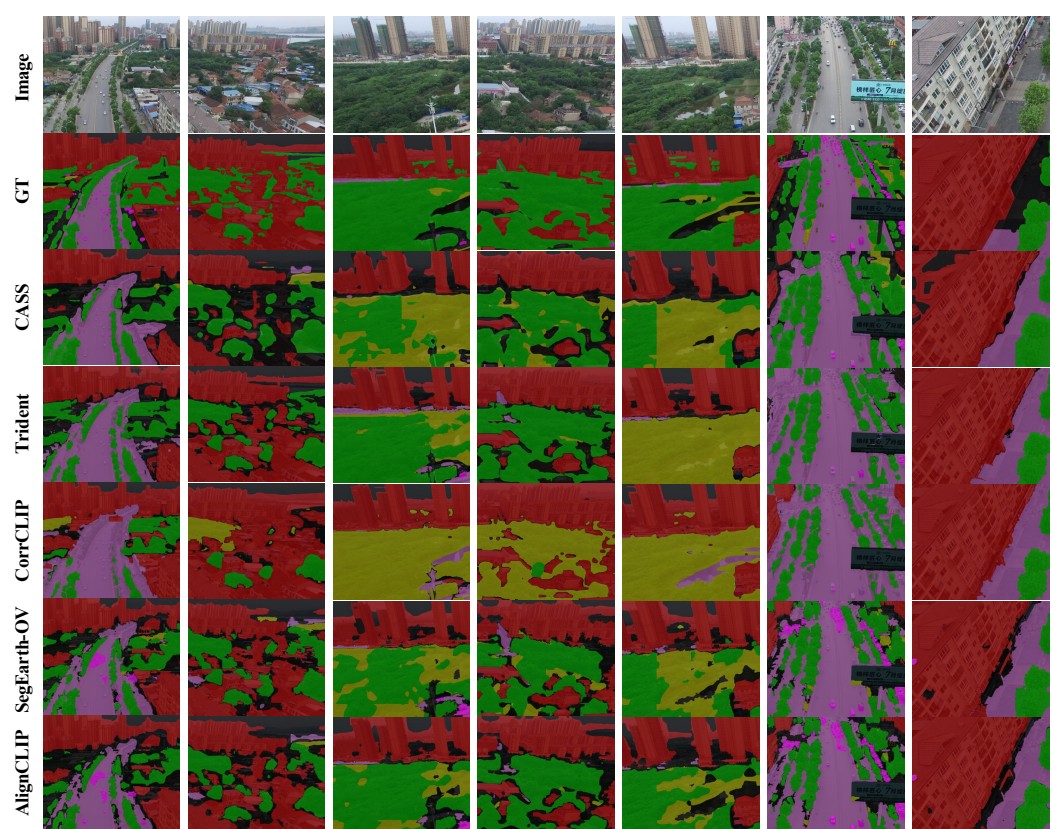

Figure 11: Qualitative comparison of different training-free OVSS methods on UAVid.

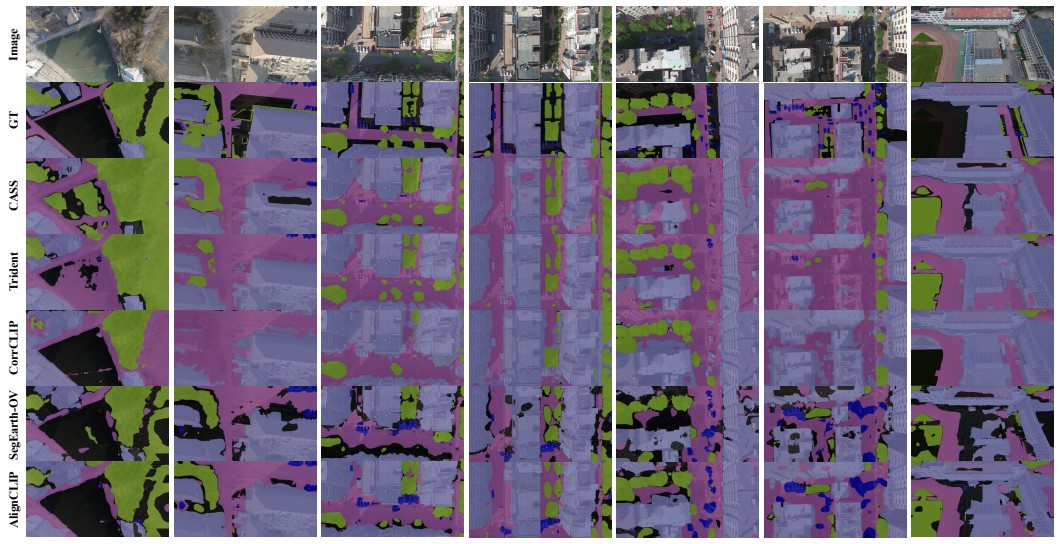

Figure 12: Qualitative comparison of different training-free OVSS methods on UDD5.

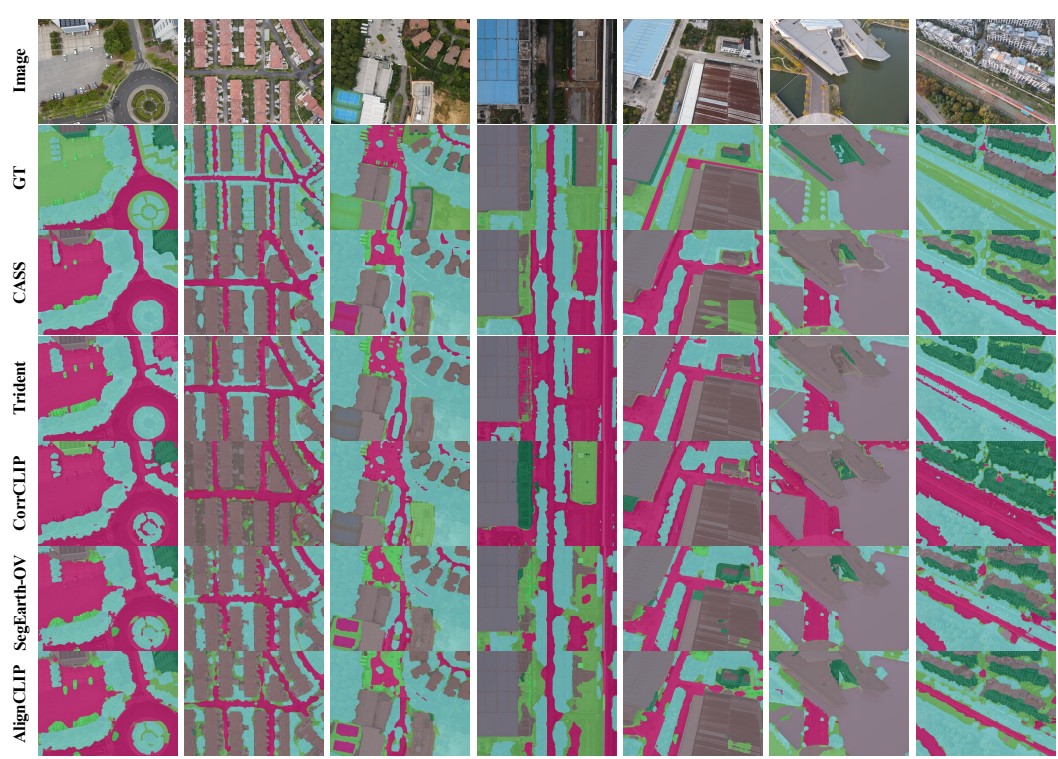

Figure 13: Qualitative comparison of different training-free OVSS methods on VDD.

