# OpenReview forum: "AlignCLIP: Self-Guided Alignment for Remote Sensing Open-Vocabulary Semantic Segmentation"
_ICLR.cc/2026/Conference — ICLR 2026 Conference Desk Rejected Submission_

### Official Review · Reviewer_QuDJ · 2025-10-25

**Soundness:** 3
**Presentation:** 4
**Contribution:** 2
**Rating:** 6
**Confidence:** 4

**Summary:**

This paper proposes AlignCLIP, a fully training-free framework for open-vocabulary semantic segmentation (OVSS) in remote sensing imagery. Observing that remote sensing objects of the same category exhibit compact intra-class feature distributions, the authors design two lightweight modules to address cross-modal mismatches between image patches and textual embeddings. The Self-Guided Alignment (SGA) module refines text embeddings using the most reliable image-specific visual prototypes, while the Cluster-Constrained Enhancement (CCE) module groups semantically similar patches, suppresses inter-cluster noise, and updates the logits map via constrained propagation. Without requiring retraining or reference sets, AlignCLIP integrates seamlessly with SegEarth-OV’s upsampling module and achieves state-of-the-art performance across eight benchmark datasets, outperforming prior training-free methods by an average of +2.2 mIoU, with strong generalization and interpretability supported by extensive ablation and qualitative analyses.

**Strengths:**

1. The motivation is clearly defined and well matched to the proposed SGA design; the paper convincingly demonstrates how the method directly addresses the identified issue.
2. Extensive experiments across multiple benchmarks show consistent performance improvements, supporting the technical soundness of the approach.
3. The adaptation to remote sensing imagery is well justified; using in-image visual references is an effective and elegant idea that leverages the unique characteristics of this domain.
4. The paper is well-organized, with a logical flow and clear explanations that make the technical ideas easy to follow.

**Weaknesses:**

1. The CCE module leverages visual feature maps and self-attention matrices from vision foundation models to constrain clustering, but similar strategies have already been widely explored in prior works. Moreover, its design does not closely connect to the paper’s main motivation of alleviating cross-modal mismatches.
2. Although the SGA module aligns well with the paper’s motivation, its individual improvement over the baseline is relatively small. The major performance gain (+2.2 mIoU) is observed only when SGA and CCE are combined, which somewhat weakens the claim of SGA’s independent effectiveness.

**Questions:**

1. In Table 2, both baseline + SGA and baseline + CCE yield very limited improvements, while the combination of SGA + CCE provides a clear gain. How do the authors explain this apparent interdependence between the two modules?
2. The paper does not provide any comparison or discussion of computational complexity or inference time. Could the authors quantify the additional cost introduced by CCE (e.g., FLOPs, runtime, or memory)?

---

> ### Author Response · Authors · 2025-11-20
> **Response to Reviewer QuDJ**
>
> Dear reviewer QuDJ,
>
> Thank you for your insightful suggestions, which are very important for improving our work. Below, we provide our responses.
>
> ## **W1: The novelty of CCE module is limited**
> > The CCE module leverages visual feature maps and self-attention matrices from vision foundation models to constrain clustering, but similar strategies have already been widely explored in prior works.
>
> Thank you for raising the concern regarding the novelty of our modules. We fully understand and value this point. We have already provided a comprehensive clarification in **General Response Part 2 (Q1)**.
>
> Furthermore, as demonstrated in our ablation studies, employing the CCE module alone yields only marginal performance gains. This observation indicates that directly transferring existing clustering-based refinement methods is insufficient to achieve robust improvements in cross-modal settings. The fundamental reason lies in the fact that the original logits map contains a substantial amount of noisy and inconsistent predictions, causing the optimization propagation of CCE to be easily disrupted and thus limiting its effectiveness.
>
> Motivated by this limitation, we further analyzed the unique property of high intra-class compactness in remote sensing imagery and introduced the SGA module to significantly improve the alignment quality of the logits map, making the predictions more stable and semantically coherent. Only after the logits map is properly corrected by SGA can the CCE module fully unleash its potential and deliver substantial performance gains.
>
> Therefore, we emphasize that the contribution of CCE is not isolated from the core motivation of the paper. On the contrary, it is precisely because SGA is designed to address the cross-modal mismatch problem that CCE becomes truly effective.
>
> ## **W2: The performance gain of SGA module is limited**
> > Its individual improvement over the baseline is relatively small. The major performance gain (+2.2 mIoU) is observed only when SGA and CCE are combined, which somewhat weakens the claim of SGA’s independent effectiveness.
>
> We appreciate the reviewer's attention to the independent performance of the SGA module. The core goal of SGA is to alleviate cross-modal mismatches, making logits map predictions within the same class more consistent and aligned. As a result, SGA can independently provide a stable improvement of approximately **+1% mIoU**, demonstrating its substantive effect on cross-modal alignment.
>
> However, SGA's effectiveness is still constrained by CLIP's limited spatial awareness, which motivated the introduction of the CCE module. CCE performs optimized propagation on logits maps aligned by SGA, yielding significantly higher gains than operating on unaligned baseline logits (up to **+2.2% mIoU**). This further validates the effectiveness of SGA: it not only independently improves prediction quality but also provides cleaner and more reliable inputs for CCE, thereby amplifying the overall performance improvement.
>
> We provide a systematic discussion of this relationship in the module ablation analysis (see revised PDF, lines 436–445). Overall, SGA's contribution is not only reflected in its independent performance gains but also in its role as a key pre-processing step that decisively enhances the overall framework performance.
>
> ## **Q1: The complementarity between SGA module and CCE module**
> > How do the authors explain this apparent interdependence between the two modules?
>
> As discussed in our response to **W2**, the complementarity between SGA and CCE arises from their natural functional alignment. The optimization and propagation mechanism of CCE heavily relies on the class consistency and alignment of the input logits map. When CCE is used alone, the original logits map contains many inaccurate or inconsistent predictions, which are easily affected by noise and limit performance gains.
>
> In contrast, SGA effectively mitigates cross-modal mismatches, generating more consistent and aligned logits distributions for each class. When CCE operates on these high-quality logits, its propagation can significantly amplify the benefits. Thus, while individual improvements from SGA or CCE alone are limited, their combination is highly complementary: SGA provides reliable inputs, and CCE further magnifies the alignment gains, resulting in a synergistic effect.
>
> ## **Q2: Efficiency analysis**
> > The paper does not provide any comparison or discussion of computational complexity or inference time. Could the authors quantify the additional cost introduced by CCE (e.g., FLOPs, runtime, or memory)?
>
> We appreciate the reviewer's attention to efficiency. We have provided a detailed efficiency analysis in the **General Response Part 3 (Q3)**.
>
> Thank you again for your suggestions on our work. We hope our Rebuttal can address your concerns, and we look forward to your reply.

---

> ### Author Response · Authors · 2025-11-27
> **Response to Reviewer QuDJ**
>
> Dear reviewer QuDJ,
>
> We hope this message finds you well. As the discussion period will conclude in a week, we would like to ensure that our responses have satisfactorily addressed your concerns. If there are any additional points or feedback you would like us to consider. Please feel free to let us know. Your feedback has been invaluable in improving the quality of our work, and we remain committed to addressing any outstanding issues. Thank you for your careful review of our work.
>
> Sincerely,
>
> The Authors

---

### Official Review · Reviewer_Cg66 · 2025-10-26

**Soundness:** 3
**Presentation:** 3
**Contribution:** 2
**Rating:** 4
**Confidence:** 3

**Summary:**

This paper addresses open-vocabulary semantic segmentation (OVSS) in remote sensing imagery with a training-free framework, AlignCLIP. The method mitigates patch–text mismatches in CLIP-based models via two modules: Self-Guided Alignment (SGA), which refines text embeddings using visual prototypes, and Cluster-Constrained Enhancement (CCE), which clusters semantically similar patch features. AlignCLIP requires no retraining or reference sets. Experiments on eight benchmarks show improved mIoU over existing training-free OVSS baselines, supported by ablation and qualitative analyses.

**Strengths:**

1.	Two distinct modules, SGA and CCE, creatively harness domain-specific characteristics for improved patch-text alignment and alleviate disturbance between irrelevant patches.
2.	Results on eight diverse remote sensing datasets present a consistent mIoU gain of +2.2% on average over the most competitive baseline (SegEarth-OV), setting new standards for training-free OVSS.
3.	Ablation studies dissected the contributions of each module and explored sensitivity to key hyperparameters.
4.	SGA is demonstrated as a plug-in for existing frameworks (Table 9), adding evidence for its generality.

**Weaknesses:**

1.	The approach is motivated by the highly compact intra-class feature distribution of remote sensing image (Line 85-88, Figure 1c), but this property is neither **empirically quantified** nor **qualified**. Additionally, there is insufficient exploration of generalization limits: will SGA/CCE degrade if intra-class variance increases (as in higher-resolution urban datasets) or with more ambiguous categories?
2.	While the paper provides clear formulations for SGA and CCE, the mathematical treatment of several key steps appears limited, and some design choices seem primarily empirically motivated rather than theoretically grounded. For instance, the claim in Lines 238–240 is not explicitly demonstrated. In particular, Section 3.3 outlines the affinity matrix masking and constraint propagation, but does not provide a detailed justification for the optimality of the binary cluster mask or a clear explanation of how propagation enhances alignment. A more in-depth discussion of how these design decisions affect intra- and inter-class confusion would further strengthen the theoretical foundation of the approach.
3.	Evaluations of reference-set-based methods would be beneficial, at least in the appendix. While the authors disable post-processing (e.g., denseCRF) to isolate method effects, the paper does not discuss how this choice impacts real-world applicability or whether simple post-processing could further narrow the gap with evaluated baselines (or reference-based methods).
4.	The conclusion in Line 357-358 is not justified.
5.	No analysis of runtime, computational cost, or efficiency is provided, which is particularly relevant for a training-free framework where clustering and affinity computation may become nontrivial for large-scale imagery.
6.	While Figure 2 gives a good high-level view of the two modules, it does not expose the detailed mechanics of cluster refinement and “Constrain” in CCE module, which leads to ambiguity.
7.	Qualitative results are visually compelling, but the absences of original image, GT label, and standardized metric for qualitative result make the conclusions somewhat confused.

**Questions:**

1.	Can the authors provide empirically quantified or qualified demonstrations on the claim in Line 85-88? (Weakness 1).
2.	Can the authors provide more rigorous theoretical justification of several key steps and design choices in Section 3.2 and Section 3.3? (Weakness 2)
3.	Relevant experiments are appreciated. (Weakness 3 & 5)
4.	Some claims require further clarification, and several minor presentation issues could be improved for clarity and consistency. (Weakness 4 & 6 & 7)

---

> ### Author Response · Authors · 2025-11-20
> **Response to Reviewer Cg66 (Part 1)**
>
> Dear reviewer Cg66,
>
> Thank you for your thoughtful insights on our paper. We hope the following responses will address your concerns.
>
> ## **W1 & Q1: Analysis of our motivation**
> > Can the authors provide empirically quantified or qualified demonstrations on the claim in Line 85-88?
>
> We appreciate the reviewer's insightful comments on the motivation of our method and have provided a detailed response in the **General Response Part 2 (Q1)**.
>
> ## **W2 & Q2: The explanation of the method is limited**
> > Can the authors provide more rigorous theoretical justification of several key steps and design choices in Section 3.2 and Section 3.3?
>
> We appreciate the reviewer's insightful comments on the method. We fully agree that providing additional theoretical explanations can enhance the interpretability and rationale of our design. To address this, we have expanded the description of key steps in the revised PDF, including the following points:
> 1. In Section 3.2 (revised PDF, lines 240–246), we provide a **detailed explanation of using visual prototypes to refine text features**: we select the visual feature most similar to the text as the visual prototype and fuse it via the SGA module. Since intra-class features in remote sensing images are highly compact (see General Response Part 2 (Q1) for detailed validation), other patch-level visual features close to the prototype are likely to belong to the same class. Therefore, by similarity-weighted fusion of the visual prototype and text features, the cross-modal mismatches can be effectively reduced, resulting in aligned text features that are more stable and discriminative for subsequent matching.
> 2. In Section 3.3 (revised PDF, lines 270–277), we provide a **detailed explanation of the binary cluster mask**: directly using the original affinity matrix obtained from VFM may introduce inter-class interference, as attention between different semantic classes is often non-zero. To avoid such undesirable inter-class interactions, we mask the affinity matrix based on clustering results, retaining only the connections between patches predicted to belong to the same cluster and setting connections across different clusters to zero. This ensures that logits propagation occurs solely within classes, preventing semantic contamination and improving prediction stability.
> 3. In Section 3.3 (revised PDF, lines 289–292), we provide a **detailed explanation of how the propagation mechanism enhances semantic consistency**: within the same cluster, logits maps are weighted and averaged based on affinity matrix, ensuring consistent predictions within the cluster while preventing interference from other clusters. This intra-cluster propagation effectively reduces intra-class variation and suppresses inter-class confusion, resulting in more accurate and stable segmentation.
> 4. We have corrected the inaccurate description in lines 238–240 of the original manuscript.
>
> ## **W3 & Q3: Lack of reference set evaluation & comparison with post-processing**
> > Evaluations of reference-set-based methods would be beneficial, While the authors disable post-processing (*e.g.*, denseCRF) to isolate method effects, the paper does not discuss how this choice impacts real-world applicability.
>
> We appreciate the reviewer's constructive comment and fully agree that a more systematic evaluation of reference-set-based methods would strengthen our work. However, as noted in the revised PDF (lines 355–358), the performance of such methods heavily depends on the construction quality of the reference set (*e.g.*, size, annotation accuracy, scene diversity), making fair comparisons under a unified setting extremely challenging. Moreover, the relevant works have not been open-sourced, preventing reliable reproduction of their reference set construction and full methodological details. Once the implementations are released, we will promptly include comprehensive experiments in the manuscript.
>
> For comparisons involving post-processing methods, we have added relevant experiments in Appendix A.8 of the revised PDF and included the comparison results, as shown in the table below:
>
> | Methods   | OpenEarthMap | LoveDA | iSAID | Potsdam | Vaihingen | UAVid | UDD5 | VDD  | Avg.  |
> |-----------|--------------|--------|-------|---------|-----------|-------|------|------|-------|
> | Ours      | 40.1         | 39.5   | 23.6  | 47.9    | 34.5      | 44.4  | 51.8 | 48.4 | 41.3  |
> | +denseCRF | 40.9         | 40.2   | 24.3  | 48.7    | 35.1      | 45.3  | 52.9 | 49.9 | 42.2 |
>
> Post-processing techniques typically refine logits maps using low-level visual cues such as color consistency and spatial continuity, providing performance gains in many scenarios. Our experiments show that denseCRF-based post-processing yields an additional **0.9%** improvement on our method. This indicates that our method is compatible with post-processing techniques, and its primary performance gains stem from our core modules rather than post-processing.

---

> ### Author Response · Authors · 2025-11-20
> **Response to Reviewer Cg66 (Part 2)**
>
> ## **W4 & Q4: conclusion is not justified**
> > The conclusion in Line 357-358 is not justified.
>
> Thank you for pointing this out, and we have corrected the inaccurate statements in the manuscript.
>
> ## **W5 & Q3: efficiency analysis**
> > No analysis of runtime, computational cost, or efficiency is provided.
>
> We appreciate the reviewer's suggestion and have conducted a comprehensive efficiency analysis in the **General Response Part 3 (Q3)**.
>
> ## **W6 & Q4: Lack of detailed mechanics of CCE module**
> > While Figure 2 gives a good high-level view of the two modules, it does not expose the detailed mechanics of cluster refinement and "Constrain" in CCE module.
>
> We appreciate the reviewer's attention to the details of the CCE module. We have provided a more comprehensive description in the revised PDF to eliminate potential ambiguities.
>
> In the CCE module, we first extract dense visual features from the image using VFM and perform clustering based on these features to obtain semantically consistent patch-level cluster assignments. This groups semantically related and visually similar regions into the same cluster, providing a structured semantic prior for subsequent constrained propagation.
>
> Next, we apply a "**constrain**" operation to the original affinity matrix from VFM:
> 1. Retain affinity values for patches within the same cluster.
> 2. Set affinity values between patches of different clusters to zero.
>
> This masking ensures that logits propagation occurs strictly within the same semantic cluster, preventing inter-cluster interference. Finally, by performing weighted propagation of the logits map over the masked affinity, we achieve more consistent and stable intra-cluster predictions while effectively suppressing inter-class leakage, thereby improving overall segmentation quality.
>
> ## **W7 & Q4: The qualitative comparison results are confused**
> > Qualitative results are visually compelling, but the absences of original image, GT label, and standardized metric for qualitative result make the conclusions somewhat confused.
>
> We appreciate the reviewer's valuable suggestion. We have updated the qualitative analysis figures (see Figure 3 and Appendix A.11 in the revised PDF), showing the original images, GT, and comparisons with relevant methods, which we hope addresses your concerns.
>
> Thank you for your valuable suggestions, which are very helpful for improving our work. We hope our responses adequately address your concerns.

---

> ### Author Response · Authors · 2025-11-27
> **Response to Reviewer Cg66**
>
> Dear reviewer Cg66,
>
> We hope you are well. With only one week left in the discussion stage, we want to ensure that our responses have resolved all of your concerns. If there is anything further you would like us to clarify or expand upon, please let us know. We sincerely appreciate your constructive feedback, which has greatly helped us refine our work. Thank you again for your thoughtful review.
>
> Sincerely,
>
> The Authors

---

### Official Review · Reviewer_hDE6 · 2025-10-27

**Soundness:** 3
**Presentation:** 3
**Contribution:** 3
**Rating:** 6
**Confidence:** 4

**Summary:**

This paper proposes **AlignCLIP**, a **training-free** framework for **open-vocabulary semantic segmentation (OVSS)** in remote sensing imagery. The authors observe that intra-class visual features in remote sensing are more compact than those in natural images, which allows representative image prototypes to refine textual embeddings. Based on this insight, the method introduces two core modules:
1. **Self-Guided Alignment (SGA)** – aligns text embeddings with image-specific visual prototypes to reduce cross-modal mismatches.
2. **Cluster-Constrained Enhancement (CCE)** – clusters semantically similar patches and propagates logits under cluster constraints to suppress irrelevant correlations.

The proposed framework requires no training or external reference sets and achieves consistent improvements over 12 state-of-the-art training-free OVSS baselines across eight remote sensing benchmarks.

**Strengths:**

- **Novel and effective idea:** The SGA module elegantly leverages image-specific prototypes for text refinement, mitigating cross-modal mismatches in a self-guided, training-free manner.
- **Strong empirical results:** Consistently outperforms recent CLIP-based training-free methods across eight diverse remote sensing datasets.
- **Motivation well grounded:** The observation about compact intra-class feature distributions in remote sensing is insightful and aligns with the method’s design.

**Weaknesses:**

- **Limited conceptual novelty:** While effective, the SGA and CCE modules are primarily heuristic improvements on existing cross-modal matching and clustering strategies.
- **No theoretical formulation:** The “alignment” process is empirically justified but lacks formal theoretical grounding or interpretability analysis.
- **Missing cross-domain validation:** The claimed generalization would be more convincing with results on non-remote-sensing datasets (e.g., ADE20K, COCO-Stuff).
- **Computation not discussed:** The clustering process in CCE may add inference overhead; runtime or complexity comparisons with SegEarth-OV are missing.
- **Writing issues:** The paper is generally clear but occasionally verbose; the notation in Eq. (2) and Sec. 3.3 could be more concise and precise.

**Questions:**

1. How sensitive is the method to the backbone choice (DINO vs. SAM) in both performance and efficiency?
2. Can the “compact intra-class feature” assumption be quantitatively analyzed (e.g., variance comparison between RS and natural image features)?
3. How does the SGA perform when intra-class diversity is higher (e.g., across seasons or sensors)?
4. How does AlignCLIP handle abstract unseen categories (e.g., “industrial area” vs. “building”)?
5. Please provide inference speed or FLOPs comparison with SegEarth-OV to support the claim of efficiency.

---

> ### Author Response · Authors · 2025-11-20
> **Response to Reviewer hDE6 (Part 1)**
>
> Dear reviewer hDE6,
>
> Thank you for your insightful suggestions, which are very helpful for improving our work.
>
> ## **W1: Limited conceptual novelty**
> > Limited conceptual novelty: While effective, the SGA and CCE modules are primarily heuristic improvements on existing cross-modal matching and clustering strategies.
>
> First, we sincerely appreciate the reviewer's recognition of the effectiveness of our method. Regarding your concern about the "limited conceptual novelty", we have provided a detailed explanation in the **General Response Part 2 (Q2)**.
>
> ## **W2: No theoretical formulation**
> > No theoretical formulation: The “alignment” process is empirically justified but lacks formal theoretical grounding or interpretability analysis.
>
> We fully understand and appreciate your concern regarding the lack of theoretical explanation. To address this, we have added an interpretability analysis of the alignment mechanism in the SGA module in the revised PDF (lines 240–246). Specifically, we first compute the similarity between visual and textual features and select, from the visual feature map, the feature most similar to each textual embedding as a visual prototype. Given our observation that remote sensing images exhibit highly compact intra-class feature distributions, these visual prototypes naturally cluster with other visual features of the same category in the feature space.
>
> We further fuse each textual embedding with its corresponding visual prototype via a similarity-based weighting strategy, effectively reducing their discrepancy in the feature space. The fused textual embeddings become more stable and more accurately aligned with the associated visual features. We believe that this alignment mechanism offers clear interpretability for SGA and provides theoretical support for the soundness of our work.
>
> ## **W3: Missing cross-domain validation**
> > Missing cross-domain validation: The claimed generalization would be more convincing with results on non-remote-sensing datasets (e.g., ADE20K, COCO-Stuff).
>
> We fully agree with your suggestion regarding cross-domain validation. Accordingly, we have added cross-domain experimental results in Section A.7 of the revised PDF, as shown in the table below:
>
> | Methods   | Cityscapes | ADE20k | Stuff | Context59 | VOC20 | Avg. |
> |-----------|------------|--------|-------|-----------|-------|------|
> | SC-CLIP   | 41.0       | 20.1   | 26.6  | 40.1      | 84.3  | 42.4 |
> | +SGA      | 38.5       | 20.0   | 26.4  | 40.0      | 77.9  | 40.6 |
> | Trident   | 42.9       | 21.9   | 28.3  | 42.2      | 84.5  | 44.0 |
> | +SGA      | 38.9       | 21.1   | 27.7  | 42.1      | 82.0  | 42.4 |
> | CorrCLIP  | 49.9       | 26.9   | 31.6  | 48.8      | 88.8  | 49.2 |
> | +SGA      | 47.8       | 26.1   | 31.1  | 47.2      | 85.8  | 47.6 |
>
> Specifically, we selected five representative natural image semantic segmentation datasets and integrated the SGA module into three SOTA methods for natural image segmentation. The results show a performance decline across all methods and datasets.
>
> It is important to note that this decline is not due to insufficient generalization of our method. Rather, it arises because intra-class feature distributions in natural images are far more dispersed than in remote sensing images, which inherently conflicts with the core motivation of our method. We have discussed this in General Response Part 2 (Q1).
>
> ## **W4 & Q5: Computation not discussed**
> > Computation not discussed: The clustering process in CCE may add inference overhead; runtime or complexity comparisons with SegEarth-OV are missing.
>
> We appreciate your valuable comment. Regarding the potential inference overhead introduced by the CCE module, we have provided a detailed discussion in the **General Response Part 3 (Q3)**.
>
> ## **W5: Writing issues**
> > Writing issues: The paper is generally clear but occasionally verbose; the notation in Eq. (2) and Sec. 3.3 could be more concise and precise.
>
> We appreciate your detailed guidance on our manuscript. We have streamlined certain formulas and descriptions in the revised PDF (see Equations 9 and 12). Regarding Equation 2, it describes performing self-attention separately on the query, key, and value matrices and computing the average feature, which is consistent with Equation 10 in SegEarth-OV.

---

> ### Author Response · Authors · 2025-11-20
> **Response to Reviewer hDE6 (Part 2)**
>
> ## **Q1: Sensitivity analysis of different backbone models**
> > How sensitive is the method to the backbone choice (DINO vs. SAM) in both performance and efficiency?
>
> We appreciate your attention to this issue. In fact, we have systematically analyzed the impact of different backbones (*e.g.*, DINO and SAM) on model performance in the appendix (see Section A.4 of the revised PDF). The experiments show that under different model architectures and hyperparameter settings, the performance of our method remains stable, indicating that it is not sensitive to backbone choice.
>
> Furthermore, as discussed in our General Response, we have added efficiency comparisons for the two model variants. Both inference speed and memory usage show similar increases across backbones, further demonstrating that our method maintains consistent efficiency characteristics regardless of the backbone.
>
> ## **Q2: Quantitative analysis of compactness of intra-class features**
> > Can the “compact intra-class feature” assumption be quantitatively analyzed (e.g., variance comparison between RS and natural image features)?
>
> We appreciate the reviewer's insightful question, which is important for strengthening the credibility and rigor of our motivation. We have provided a comprehensive analysis of this issue in the **General Response Part 2 (Q1)**.
>
> ## **Q3: The impact of intra-class diversity**
> > How does the SGA perform when intra-class diversity is higher (e.g., across seasons or sensors)?
>
> We appreciate your attention to this issue. As discussed in the introduction, the SGA module refines text features by selecting the visual prototype most similar to the text. Since intra-class features in remote sensing images are highly compact in feature space, other features close to the prototype are likely to belong to the same class. Aligning text features with the visual prototype thus significantly improves the stability and accuracy of cross-modal matching, which is a key reason for SGA's effectiveness in remote sensing scenarios.
>
> However, when intra-class diversity increases substantially (*e.g.*, across seasons, sensors, or natural image data), the distribution of same-class features becomes more dispersed (see revised PDF, lines 85–99). This high dispersion contradicts the core assumption of our method and diminishes the benefits of visual prototype alignment. Additional experiments in **W3** further confirm that SGA's effectiveness decreases under such dispersed intra-class distributions.
>
> ## **Q4: abstract and unseen categories**
> > How does AlignCLIP handle abstract unseen categories (e.g., “industrial area” vs. “building”)?
>
> We appreciate your attention to this issue. Our work focuses on open-vocabulary semantic segmentation, so the model’s ability to handle abstract unseen categories (e.g., industrial area vs. building) primarily depends on the open-vocabulary recognition capability of the underlying cross-modal model.
>
> In this study, we use CLIP B/16 as the visual–text alignment backbone. Benefiting from its training on large-scale image–text contrastive data, CLIP exhibits strong generalization to unseen categories and abstract semantic concepts. Thus, even if these categories are not labeled in the datasets, CLIP can provide distinguishable text embeddings and visual matches, enabling AlignCLIP to effectively recognize and handle such unseen categories.
>
> Thank you for carefully reviewing our work. We look forward to your feedback on any further questions.

---

> ### Author Response · Authors · 2025-11-27
> **Response to Reviewer hDE6**
>
> Dear reviewer hDE6,
>
> We hope this message finds you well. As the discussion period will end in one week, we would like to confirm that all your concerns have been adequately addressed. Please let us know if there are any remaining issues or additional feedback. Your comments have been instrumental in improving our work. Thank you for your time and consideration.
>
> Sincerely,
>
> The Authors

---

> > ### Comment · Reviewer_hDE6 · 2025-11-28
> >
> > Thank you for your response. While most of my concerns have been addressed, I still have some reservations regarding the originality and novelty of the newly proposed modules. Therefore, I intend to maintain my current positive scores for now and will finalize my recommendation after considering the feedback from other reviewers.

---

> > > ### Author Response · Authors · 2025-11-28
> > > **Response to Reviewer hDE6**
> > >
> > > Dear reviewer hDE6,
> > >
> > > Thank you for your additional feedback. We are glad to hear that many of your concerns have been addressed, and we appreciate the opportunity to further clarify the remaining issues. We will do our utmost to resolve your concerns.
> > >
> > > In the **training-free open-vocabulary semantic segmentation setting**, most existing methods primarily focus on **modifying the attention mechanisms of the CLIP visual encoder** to obtain more spatially discriminative visual features. However, these approaches tend to overlook a crucial issue: **the significant mismatch that still exists between patch-level visual features and text features**.
> > >
> > >
> > > Our work approaches the problem from this underexplored perspective by proposing to **refine the text features** themselves to mitigate cross-modal mismatches—a design philosophy that differs substantially from existing methods.
> > >
> > > Based on our systematic observations and quantitative analyses in remote sensing scenarios, we verify the key property that "**intra-class visual features in remote sensing images are much more compact.**" Building on this insight, we design a **simple yet effective alignment module** to enhance the cross-modal alignment capability of text embeddings.
> > >
> > > Specifically, we select the most reliable visual prototype within each class using cosine similarity and align the text embedding with this prototype, making the text representation closer to the true visual semantics in the feature space. Due to the high intra-class compactness in remote sensing imagery, features that are close to the visual prototype are very likely to belong to the same category. Consequently, the aligned text embedding can more accurately and consistently match other same-class visual features, thereby significantly improving cross-modal matches.
> > >
> > > Moreover, our method is not only simple and lightweight but also **consistently effective across different methodological paradigms**. In the experiments presented in Appendix A.5, we integrate the SGA module into three representative training-free OVSS frameworks and observe consistent improvements: +1.7% on ProxyCLIP, +1.9% on SC-CLIP, and +1.6% on CorrCLIP. These results demonstrate that our module can deliver tangible gains with minimal modifications, highlighting its **independent contribution and practical value**.
> > >
> > > | Methods   | OpenEarthMap | LoveDA | iSAID | Potsdam | Vaihingen | UAVid | UDD5 | VDD  | Avg.   |
> > > |:--------:|:------------:|:------:|:-----:|:-------:|:---------:|:-----:|:----:|:----:|:----------:|
> > > | ProxyCLIP | 35.0         | 33.5   | 20.7  | 44.1    | 27.8      | 42.1  | 46.5 | 44.3 | 36.8   |
> > > | +SGA  | **38.6**     | **34.2**   | **21.6**  | **44.6**    | **32.7**      | **42.4**  | **48.3** | **45.3** | **38.5** |
> > > | SC-CLIP   | 35.9         | 31.7   | 18.4  | 43.4    | 29.6      | 38.3  | 42.0 | 41.0 | 35.0   |
> > > | +SGA  | **39.8**     | **32.8**   | **19.6**  | **43.6**    | **31.6**      | **39.6**  | **46.0** | **42.3** | **36.9** |
> > > | CorrCLIP  | 35.4         | 32.7   | 16.9  | 42.6    | 24.7      | 38.1  | 40.1 | 37.7 | 33.5   |
> > > | +SGA  | **36.6**     | **33.5**   | **18.6**  | **43.8**    | **27.9**      | **39.9**  | **41.1** | **39.6** | **35.1** |
> > >
> > > We hope the above clarifications help alleviate your concerns regarding the novelty of our method.
> > >
> > > Once again, we sincerely appreciate your continued attention and valuable feedback on our work.
> > >
> > > Sincerely,
> > >
> > > The Authors

---

### Official Review · Reviewer_pRf8 · 2025-10-31

**Soundness:** 2
**Presentation:** 3
**Contribution:** 3
**Rating:** 4
**Confidence:** 3

**Summary:**

This paper presents AlignCLIP, a training-free framework designed for Open-Vocabulary Semantic Segmentation (OVSS) in remote sensing imagery. The study is motivated by two key observations. First, directly applying CLIP to remote sensing imagery typically results in cross-modal mismatches between visual and textual features. Second, objects belonging to the same category in remote sensing images often exhibit more compact intra-class feature distributions compared to natural images, offering opportunities for improved alignment. Based on these insights, the authors propose two modules: a Self-Guided Alignment (SGA) module, which adaptively aligns textual embeddings with representative visual prototypes extracted from image patches, and a Cluster-Constrained Enhancement (CCE) module, which clusters semantically similar patches to reinforce intra-cluster consistency and improve segmentation quality. Experiments conducted on eight remote sensing benchmarks demonstrate that AlignCLIP achieves state-of-the-art performance. The authors further validate the generalization capability of AlignCLIP by integrating the proposed modules into other OVSS frameworks.

**Strengths:**

* **Originality:**

  This paper addresses a significant and challenging task in remote sensing: open-vocabulary semantic segmentation without additional training. The proposed AlignCLIP framework demonstrates originality through its combination of image-specific visual prototypes and self-guided alignment (SGA) mechanisms. This novel strategy circumvents reliance on external datasets or training, effectively extending the cross-modal semantic alignment capabilities of CLIP-based methods to remote sensing scenarios.
* **Quality:**

  The quality of the research is solid, demonstrated by extensive experimental validation on eight standard remote sensing datasets. The quantitative comparisons indicate that AlignCLIP consistently achieves competitive performance improvements. Additionally, the ablation studies effectively analyze both individual components and hyperparameter settings, clearly highlighting the effectiveness of the proposed SGA and CCE modules.
* **Clarity:**

  In terms of clarity, the paper is generally well-organized and logically structured. Methodological details are systematically described, and figures such as Figure 2 provide intuitive visualizations that help readers quickly grasp the core concepts. The pseudo-code included in the appendix further enhances reproducibility and reader comprehension.
* **Significance:**

  Significance arises from both methodological innovation and practical applicability. AlignCLIP provides a robust and adaptable foundation module that can seamlessly integrate into existing CLIP-based methods. This adaptability substantially broadens its potential impact within the research community and practical applications.

**Weaknesses:**

### Motivation
* **Feature Distribution Comparison:**

  In Figure 1, the authors suggest that objects belonging to the same category exhibit more compact feature distributions in remote sensing imagery compared to natural images. However, this conclusion warrants reconsideration. Specifically, the visualized points in the natural image example represent distinct semantic categories ("person" and "clothes"), inherently leading to greater semantic separation than two instances of "trees" in remote sensing imagery. Additionally, prior research, such as Figure 7 in DeCLIP [1], demonstrates that even different parts of the same object (e.g., a dog's ear vs. its body) are distinctly separated in feature space when using VFMs or VFM-enhanced CLIP.

  If the authors intend to highlight that natural images typically involve finer-grained semantic components leading to more diverse feature distributions, whereas remote sensing imagery captures more homogeneous details from greater distances, it would be beneficial to explicitly state this. Otherwise, additional clarification should be provided.

* **Cross-modal Mismatch Justification:**

  Another primary motivation discussed in the paper is the “cross-modal mismatch” between CLIP’s visual and textual encoders. However, the manuscript lacks concrete quantitative or qualitative evidence to substantiate this claim. Additionally, the cited work [2] appears irrelevant, as it utilizes only CLIP’s textual encoder, combined with a different semantic segmentation network for visual encoding. Given that cross-modal semantic alignment is generally considered a strength of CLIP, it remains unclear whether the mismatch identified originates from CLIP itself or from the self-self attention used within the proposed framework. This ambiguity weakens the foundational motivation for the approach.

### Methodology
* **Contradictory Approach:**

  The methodological description also presents confusion. If, as stated in the introduction, a cross-modal mismatch exists between CLIP’s visual and textual branches, then why does the proposed method directly leverage visual/textual attention scores to align image-specific visual prototypes? This approach appears contradictory to the stated premise and requires clarification.

### Experiments
* **Hyperparameter Sensitivity:**

  As indicated in Table 5 of the appendix, the hyperparameters $\alpha$ and $\beta$ differ substantially across datasets. Tables 6–8 further suggest high sensitivity of performance outcomes to these hyperparameters. Although the authors' transparency and thoroughness in ablation studies are commendable, this still raises concerns regarding the generalizability and robustness of the proposed method.

* **Experimental Setup Ambiguity:**

  Additionally, the experimental setting described in the manuscript is ambiguous. The authors should clearly define whether AlignCLIP is proposed as a standalone approach or as a plug-and-play module. If presented as a standalone method, an ablation study replacing SegEarth-OV's upsampling module with standard upsampling methods is necessary, as this module significantly contributes to SegEarth-OV's performance. Directly incorporating this module into AlignCLIP and comparing it against SegEarth-OV as separate approaches would thus be unfair. Conversely, if AlignCLIP is intended as a plug-and-play module, the experimental section should be restructured accordingly. The appendix data (Table 9) should be moved to Table 1 in the main text to compare the accuracy of various OVSS methods before and after applying AlignCLIP.

### Minor Issues
* The term "image-specific visual prototypes" (line 220) appears abstract and may not accurately reflect the intended meaning. Consider using "text-specific visual prototypes" if this aligns better with the authors' intended concept.
* The preliminary section (line 199) references "Following the practice of prior works" without providing specific citations.
* I suggest reviewing verb tense usage, as the manuscript currently mixes tenses, e.g., "propose" in line 90 and "designed" in line 91. Adopting consistent tenses separately when describing methods and experiments could enhance clarity.

[1] DeCLIP: Decoupled Learning for Open-Vocabulary Dense Perception

[2] Learning transferable land cover semantics for open vocabulary interactions with remote sensing images

**Questions:**

Please refer to the weaknesses section for my concerns. My initial judgment is BR, and I will adjust my rating accordingly based on the authors' response.

---

> ### Author Response · Authors · 2025-11-20
> **Response to Reviewer pRf8 (Part 1)**
>
> Dear reviewer pRf8,
>
> Thank you for your patience and thoughtful comments on our work. Below, we provide our responses to your concerns.
>
> ## **W1: Feature Distribution Comparison**
> > The authors suggest that objects belonging to the same category exhibit more compact feature distributions in remote sensing imagery compared to natural images. However, this conclusion warrants reconsideration.
>
> We appreciate the reviewer's concern regarding our motivation. We fully agree that directly comparing features from different semantic categories in natural images (*e.g.*, "person" vs. "clothes") with features of the same category in remote sensing images (*e.g.*, "trees") may indeed introduce bias. Accordingly, we have provided a detailed analysis of our motivation in the **General Response Part 2 (Q1)**.
>
> ## **W2: Cross-modal Mismatch Justification**
> > Another primary motivation discussed in the paper is the “cross-modal mismatch” between CLIP’s visual and textual encoders. However, the manuscript lacks concrete quantitative or qualitative evidence to substantiate this claim.
>
> We appreciate the reviewer's critical feedback and fully agree that the original manuscript did not sufficiently discuss cross-modal mismatches. To address this, we have added a more systematic analysis in the revised PDF (lines 53–79) and further clarify our key points here.
>
> First, we agree that in global understanding tasks such as image classification, CLIP's **image-level global features align well with text features**, which is one of its core strengths.
>
> However, since CLIP is pre-trained for contrastive learning-based image classification, its **patch-level dense features do not align well with text** in more challenging segmentation tasks. This is because patch-level features lack explicit spatial awareness, causing patch-level semantics to be easily confused when aligning with text.
>
> Our method addresses this issue by selecting patches most similar to the text as visual prototypes, guiding other mismatched visual features in the feature space to better align with text, thereby achieving more stable patch-level alignment.
>
> Additionally, we have rechecked and corrected the improper citation of [1], and we sincerely thank the reviewer for pointing this out.
>
> *[1] Zermatten V, Castillo-Navarro J, Marcos D, et al. Learning transferable land cover semantics for open vocabulary interactions with remote sensing images[J]. ISPRS Journal of Photogrammetry and Remote Sensing, 2025, 220: 621-636.*
>
> ## **W3: Contradictory Approach**
> > The methodological description also presents confusion.
>
> We appreciate the reviewer's insightful comment. We understand why there may appear to be a "contradiction", in fact, our method is fully consistent with our motivation. We clarify as follows.
>
> Although CLIP's patch-level features exhibit cross-modal mismatches, each class still contains a small number of patches—e.g., those located in the core region of the class—that have the highest semantic consistency with the text features. These patches can be considered local-optimal, low-noise, and highly cross-modally consistent visual prototypes. We only use the most reliable patches with the smallest feature gap for alignment, rather than blindly using all visual features, thus avoiding noise amplification.
>
> Once the **top-1** visual prototype is selected, SGA module aligns the text features with this prototype, guiding other mismatched features to align correctly. This process is fully consistent with our motivation, not contradictory.
>
> Moreover, as shown in the revised PDF (lines 488–502), we conducted a sensitivity analysis on the number of prototypes. The results indicate that using the **top-1** visual prototype achieves the best performance. This is because including too many patch features introduces noise from cross-modal mismatches, which can misdirect the alignment. Therefore, our method serves as a correction for cross-modal mismatches, rather than conflicting with the motivation.

---

> ### Author Response · Authors · 2025-11-20
> **Response to reviewer pRf8 (Part 2)**
>
> ## **W4: Hyperparameter Sensitivity**
> > As indicated in Table 5 of the appendix, the hyperparameters $\alpha$and $\beta$differ substantially across datasets.
>
> We fully understand the reviewer's concern regarding hyperparameter sensitivity. Indeed, hyperparameter settings vary across datasets, but as shown in our ablation studies, this mainly arises from the intrinsic properties and structural characteristics of each dataset.
>
> Take the hyperparameter $\beta$ as an example (see revised PDF, lines 462–471). This parameter is closely related to the image spatial scale. In larger-scale datasets (*e.g.*, OpenEarthMap), more local details are lost during feature extraction, which are crucial for accurately selecting visual prototypes. Therefore, stronger upsampling capability is required to supplement fine-grained information, favoring smaller $\beta$ values in such scenarios.
>
> Although hyperparameters differ across datasets, as long as they remain within a reasonable range, our method consistently outperforms baseline methods. Furthermore, modest adjustments of these hyperparameters can further maximize model performance.
>
> ## **W5: Experimental Setup Ambiguity**
> > The experimental setting described in the manuscript is ambiguous.
>
> We thank the reviewer for the detailed suggestion regarding the experimental setup. To clarify, **AlignCLIP is proposed as an independent method**. The SGA module relies solely on CLIP's visual and text features, without any specific model architecture or trainable components, ensuring good transferability. The experiments in Appendix A.5 are intended to demonstrate the plug-and-play nature and generality of SGA, rather than defining AlignCLIP itself as a "plugin", we will clarify this further in the revised version.
>
> Furthermore, regarding the comparison with SegEarth-OV, we would like to emphasize the following points:
> 1. The upsampling module we use is trained and does not fall within the training-free setting. For both SegEarth-OV and our method, upsampling serves only as a generic component.
> 2. The goal of AlignCLIP is not to replace or weaken SegEarth-OV, but to address a different key challenge in remote sensing on top of its effective structure. SegEarth-OV addresses the "small object scale" challenge in remote sensing using a trained upsampling module. Our work builds upon this validated pipeline and further introduces SGA and CCE to tackle cross-modal mismatches. The two address complementary issues, making our method an effective enhancement of a strong baseline.
>
> ## **Minor Issues**
> ### **Q1: Inaccurate terminology**
> > The term "image-specific visual prototypes" (line 220) appears abstract and may not accurately reflect the intended meaning.
>
> We appreciate the reviewer's suggestion, fully agree with your understanding, and have corrected the terminology in the revised manuscript.
>
> ### **Q2: Missing reference**
> > The preliminary section (line 199) references "Following the practice of prior works" without providing specific citations.
>
> We appreciate the reviewer's careful and patient review, and have added the missing citation in the revised manuscript.
>
> ### **Q3: Verb tense usage**
> > I suggest reviewing verb tense usage, as the manuscript currently mixes tenses, e.g., "propose" in line 90 and "designed" in line 91.
>
> We appreciate the reviewer's valuable suggestion and will thoroughly review the use of tense in the final version to meet the high standards of the ICLR community.
>
> Once again, we sincerely appreciate your constructive suggestions. If you have any further questions, please feel free to let us know.

---

> ### Author Response · Authors · 2025-11-27
> **Response to Reviewer pRf8**
>
> Dear reviewer pRf8,
>
> We hope you are doing well. With the discussion phase approaching its final week, we want to make sure that our replies have fully clarified your concerns. Should you have any further comments or points you would like us to address, we would be grateful to hear them. Your thoughtful feedback has greatly contributed to strengthening our work. Thank you again for your careful review and valuable suggestions.
>
> Sincerely,
>
> The Authors

---

### Official Review · Reviewer_dPaj · 2025-10-31

**Soundness:** 2
**Presentation:** 3
**Contribution:** 3
**Rating:** 4
**Confidence:** 3

**Summary:**

This paper introduces a training-free paradigm for open-vocabulary semantic segmentation (OVSS) in remote sensing imagery, aiming to bridge the gap between visual representations and textual semantics without requiring additional fine-tuning. The proposed framework centers on enhancing the alignment between image patches and textual features through two main modules: (1) a Self-Guided Alignment (SGA) module that refines text embeddings by leveraging self-similarity within visual-textual pairs, and (2) a Cluster-Constrained Enhancement (CCE) module that suppresses inter-cluster correlations and propagates cluster-level constraints to refine the final logits map. Experimental results across eight remote sensing datasets demonstrate that the proposed approach achieves state-of-the-art performance, outperforming both generic open-vocabulary segmentation methods and the remote sensing–specific SegEarth-OV baseline.

**Strengths:**

1.The methodology is intuitive and well-motivated, combining clustering-based feature alignment with constraint propagation in a training-free setting. The idea of incorporating self-guided refinement of text embeddings is appealing, especially for data-limited remote sensing applications.

2.The experimental results are impressive, showing consistent improvements over a wide range of datasets covering various remote sensing domains and annotation styles. This broad evaluation supports the generality and robustness of the proposed approach.

3.The paper is clearly written and easy to follow, with well-organized figures and method descriptions. The framework is presented in a coherent and reproducible manner.

**Weaknesses:**

1.While the paper claims that the approach is motivated by the observation that objects of the same category tend to exhibit a compact spatial distribution in remote sensing images, the explanation and empirical evidence for this phenomenon remain qualitative and underdeveloped. A more rigorous analysis (e.g., feature-space visualization or cluster compactness metrics) would strengthen this motivation.

2.The SGA module appears conceptually similar to existing clustering-based alignment mechanisms, and the upsampling module is directly adopted from SegEarth-OV. As a result, the overall methodological novelty feels somewhat limited, especially considering the rapid progress in open-vocabulary segmentation. The authors could clarify which parts are newly designed and how they differ algorithmically or conceptually from prior work.

3.In the comparative analysis, SegEarth-OV is the only baseline specifically tailored for remote sensing. Most other baselines are general-purpose open-vocabulary segmentation models trained on natural images. This imbalance in comparison may limit the fairness of the evaluation and the strength of the claimed superiority.

4.The performance gains across datasets vary considerably—some datasets show large improvements, while others only marginal gains. The paper does not adequately discuss potential causes of this variability (e.g., dataset scale, domain similarity, or text-label diversity), which weakens the interpretability of the results.

5.Although the approach is labeled “training-free,” there remains ambiguity regarding the extent of adaptation involved (e.g., text prompt selection, clustering hyperparameters). Clarifying whether these components require any dataset-specific tuning would make the claims more convincing.

**Questions:**

1.Please elaborate on the motivation behind the compact distribution assumption. Are there empirical studies or quantitative results that support this observation? How consistent is this phenomenon across different remote sensing scenes (urban, agricultural, oceanic, etc.)?

2.The novelty of the SGA and CCE modules should be discussed in greater depth, particularly in relation to existing clustering-based refinement or graph-based propagation mechanisms. It would be helpful to highlight what is unique in the proposed design and whether it introduces new theoretical insights or performance benefits beyond engineering modifications.

3.To strengthen the evaluation, please consider including more remote sensing–specific baselines or fine-tuned variants of existing ones, ensuring that the comparison fairly reflects the performance within the same domain.

4.The authors are encouraged to analyze the performance variability across datasets, possibly through ablation or correlation analysis between dataset characteristics (resolution, annotation type, text granularity) and performance gain.

5.It would also be beneficial to include efficiency analyses, such as runtime or computational complexity, to further justify the practicality of the proposed training-free paradigm.

---

> ### Author Response · Authors · 2025-11-20
> **Response to Reviewer dPaj**
>
> Dear reviewer dPaj,
>
> Thank you for your valuable comments, which are very helpful for improving our manuscript. Below, we provide our detailed response.
>
> ## **W1 & Q1: Analysis of our motivation**
> > Please elaborate on the motivation behind the compact distribution assumption.
>
> We thank the reviewer for the valuable comments on our motivation analysis. We have provided a comprehensive and thorough discussion of the motivation in the General **Response Part 2 (Q1)**.
>
> ## **W2 & Q2: The novelty of the method is limited**
> > The authors could clarify which parts are newly designed and how they differ algorithmically or conceptually from prior work.
>
> We fully understand the reviewer's concern regarding the novelty of our method and have provided a detailed explanation in the **General Response Part 2 (Q2)**.
>
> ## **W3 & Q3: Lack of comparison of baseline methods in remote sensing field**
> > Please consider including more remote sensing–specific baselines or fine-tuned variants of existing ones.
>
> Thank you for pointing this out. We agree that incorporating additional open-vocabulary segmentation methods in remote sensing could further validate our work.
>
> However, publicly available OVSS methods in remote sensing are extremely limited. In the rebuttal, we have added a discussion of [1], unfortunately, this work is not open-sourced, and the paper reports results only on the iSAID dataset, without reproducible experiments on other public remote sensing datasets (*e.g.*, VDD, LoveDA and UDD5). Therefore, comprehensive comparisons across more datasets are not feasible.
>
> Nonetheless, we have included the results of this work on iSAID, as shown in the table below:
>
> | Method       | iSAID |
> |--------------|-------|
> | SegEarth-OV  | 21.7  |
> | DGL-RSIS [1] | 21.6  |
> | **Ours**         | **23.6**  |
>
> It can be observed that under the same evaluation setting, our method still achieves the best performance, further demonstrating its advantages and generalization capability in open-vocabulary segmentation for remote sensing.
>
> We will continue to monitor open-source developments in the community, and as more remote sensing OVSS methods become available, we will incorporate them in comprehensive comparisons in future work.
>
> *[1] Li B, Zhang C, Timmerman R M, et al. DGL-RSIS: Decoupling Global Spatial Context and Local Class Semantics for Training-Free Remote Sensing Image Segmentation[J]. arXiv preprint arXiv:2509.00598, 2025.*
>
> ## **W4 & Q4: Performance gains across datasets vary considerably**
> > The authors are encouraged to analyze the performance variability across datasets.
>
> We appreciate the reviewer's suggestion, which helps enhance the interpretability of our work. In the revised PDF (lines 410–415), we have added a detailed analysis of performance differences across datasets.
>
> Specifically, the improvement over baselines varies across datasets. For example, on OpenEarthMap, the gain is only about 0.3%, primarily due to **Large-scale images weaken small-object features**.
>
> In OpenEarthMap, the high resolution and large spatial extent cause many instances (*e.g.*, buildings) to occupy only a few pixels, so their features are easily overwhelmed by background signals during extraction. This leads to low-quality vision–text feature alignment and suboptimal logits maps, which in turn limits the effectiveness of subsequent propagation and results in only modest overall improvements.
>
> ## **W5: Adaptability to different datasets**
> > Clarifying whether these components require any dataset-specific tuning would make the claims more convincing.
>
> Thank you for your profound insights. In Appendix A.3 of the revised PDF, we provide detailed descriptions of the hyperparameter settings and adaptation across datasets.
>
> It is important to note that although our method is training-free, we perform lightweight adjustments to a few hyperparameters to accommodate differences in image scale, class granularity, and object size distribution across datasets. These adjustments do not involve learning or fitting and do not rely on any annotated data, and thus do not violate the definition of a training-free method.
>
> ## **Q5: Efficiency analysis**
> > It would also be beneficial to include efficiency analyses, such as runtime or computational complexity, to further justify the practicality of the proposed training-free paradigm.
>
> We appreciate the reviewer's suggestion and have provided a detailed efficiency analysis  in the **General Response Part 3 (Q3)**.
>
> If you have any further concerns, please feel free to let us know. Thank you for considering our rebuttal.

---

> ### Author Response · Authors · 2025-11-27
> **Response to Reviewer dPaj**
>
> Dear reviewer dPaj,
>
> We hope this message finds you well. As the discussion period will conclude in about one week, we would like to ensure that our responses have addressed all of your concerns. If there are any additional points or feedback you would like us to consider, please feel free to let us know. Your insights have been extremely valuable in improving our work, and we are committed to resolving any remaining issues. Thank you again for the time and effort you have devoted to reviewing our work.
>
> Sincerely,
>
> The Authors

---

### Author Response · Authors · 2025-11-20
**General Response Part 1**

Dear Reviewers,

We sincerely thank the reviewers for their thoughtful and constructive feedback. We have devoted considerable effort to preparing detailed responses. As emphasized in the reviews, the main strengths of our work include:
1. Well-motivated and highly aligned with the proposed method (Reviewers $\color[RGB]{140,28,19}{\mathrm{dPaj}}$, $\color[RGB]{140,28,19}{\mathrm{hDE6}}$, $\color[RGB]{140,28,19}{\mathrm{QuDJ}}$).
2. Addresses the unique challenges of training-free segmentation in remote sensing without relying on external datasets or training (Reviewer $\color[RGB]{140,28,19}{\mathrm{pRf8}}$).
3. The SGA module effectively and elegantly mitigates cross-modal mismatches (Reviewer $\color[RGB]{140,28,19}{\mathrm{hDE6}}$).
4. The SGA module can be used as a plug-in to enhance other frameworks, which further demonstrates its generality (Reviewers $\color[RGB]{140,28,19}{\mathrm{pRf8}}$, $\color[RGB]{140,28,19}{\mathrm{Cg66}}$).

We summarize the key concerns raised by the reviewers as follows:
1. Clarification and quantitative analysis of the motivation (Reviewers $\color[RGB]{140,28,19}{\mathrm{dPaj}}$, $\color[RGB]{140,28,19}{\mathrm{pRf8}}$, $\color[RGB]{140,28,19}{\mathrm{hDE6}}$, $\color[RGB]{140,28,19}{\mathrm{Cg66}}$).
2. The novelty of the method is limited (Reviewers $\color[RGB]{140,28,19}{\mathrm{dPaj}}$, $\color[RGB]{140,28,19}{\mathrm{hDE6}}$, $\color[RGB]{140,28,19}{\mathrm{QuDJ}}$).
3. Lack of efficiency analysis (Reviewers $\color[RGB]{140,28,19}{\mathrm{dPaj}}$, $\color[RGB]{140,28,19}{\mathrm{hDE6}}$, $\color[RGB]{140,28,19}{\mathrm{Cg66}}$, $\color[RGB]{140,28,19}{\mathrm{QuDJ}}$).

We have addressed these issues comprehensively in **General Response Parts 2 and 3**, and we updated the manuscript with additional experimental results.

These comments have greatly strengthened and improved our work. Next, we will provide individual responses to each reviewer's comments, aiming to fully address your concerns.

Thank you for your thoughtful and patient feedback.

Sincerely,

The Authors

---

> ### Author Response · Authors · 2025-11-20
> **General Response Part 2**
>
> ## **Q1: Clarification and quantitative analysis of the motivation (Reviewers $\color[RGB]{140,28,19}{\mathrm{dPaj}}$, $\color[RGB]{140,28,19}{\mathrm{pRf8}}$, $\color[RGB]{140,28,19}{\mathrm{hDE6}}$, and $\color[RGB]{140,28,19}{\mathrm{Cg66}}$)**
>
> We appreciate the reviewers' thoughtful suggestion regarding our method's motivation. We fully agree that thoroughly validating the observation that "intra-class features in remote sensing images are more compact" is crucial for strengthening the reliability of our work.
>
> To this end, we first systematically explain why intra-class features in remote sensing images are more compact:
> 1. **Detail dilution due to long-distance imaging**: Remote sensing images are captured from much higher altitudes than natural images, making fine-grained structures hard to resolve. Objects appear visually more similar, reducing intra-class appearance variations.
> 2. **Single top-down viewpoint reduces diversity**: Remote sensing images are almost always captured from a top-down view, and this single viewpoint significantly reduces the feature diversity of objects within the same class in the dataset. This makes same-class objects (*e.g.*, water, rooftops, tree crowns) more uniform in shape and texture.
> 3. **Predictable color and texture distributions**: Remote sensing categories exhibit strong consistency (*e.g.*, water is blue, rooftops have regular textures, farmland shows block patterns), in stark contrast to the higher appearance diversity in natural images.
>
> Based on the above reasons, we conclude that "intra-class features in remote sensing images are more compact". Rather than relying solely on visual examples, our study also includes quantitative validation experiments (see revised PDF, lines 85–99), with results summarized in the table below:
>
> | Image domain   | #Pairs   | Similarity   |
> |----------------|----------|--------------|
> | Natural Image  | 331,998  | 0.67±0.10    |
> | Remote Sensing | 342,799  | 0.89±0.05    |
>
> Specifically, we computed the mean similarity and standard deviation of intra-class features in natural image scenes (ADE20K) and remote sensing image scenes (OpenEarthMap). The results show that intra-class feature similarity is significantly higher in remote sensing images (**0.89 vs. 0.67**) and exhibits lower variance, indicating more concentrated and stable feature distributions. This strongly supports our key hypothesis that intra-class features in remote sensing images are more compact.
>
> In Appendix A.9 of the revised PDF, we provide multi-class feature visualizations for natural and remote sensing images. The visualizations show that features in natural images are more prone to confusion, whereas features in remote sensing images are more compact and better separated across classes, providing visual support for our quantitative findings.
>
> ## **Q2: The novelty of the method is limited (Reviewers $\color[RGB]{140,28,19}{\mathrm{dPaj}}$, $\color[RGB]{140,28,19}{\mathrm{hDE6}}$, and $\color[RGB]{140,28,19}{\mathrm{QuDJ}}$)**
>
> We appreciate the reviewers' attention to the novelty of our method. We fully understand your concerns and would like to further clarify the core contributions of our work, as well as their fundamental differences from existing approaches:
> 1. The SGA module is directly motivated by the **highly compact intra-class feature distribution in remote sensing images**. To address cross-modal mismatches, we adopt a "**vision-to-text refinement**" mechanism, which goes beyond conventional clustering methods that are typically applied within a single modality (either vision or text). Existing OVSS studies primarily enhance the CLIP visual encoder, while the text modality is typically kept unchanged, and **explicit refinement of text features is rarely explored**. our work specifically fills this gap.
> 2. The CCE module introduces **clustering-constrained attention propagation**, which substantially differs from existing graph-based propagation methods. Prior methods generally rely on predefined attention maps and propagate information indiscriminately, often causing **incorrect cross-category propagation**. In contrast, CCE module constructs a **category-consistency mask** to explicitly block such cross-category propagation, representing a fundamental distinction from previous approaches.
> 3. The upsampling module is a generic, training-based component that is orthogonal to our work and **does not fall within the scope of training-free methods**. In the training-free setting, it functions solely as a standard utility and is entirely independent of our proposed cross-modal alignment mechanism. Our primary contribution lies in addressing the widely overlooked issue of cross-modal mismatches in training-free open-vocabulary segmentation.
>
> Our method effectively addresses a key deficiency of existing approaches—cross-modal mismatches. These innovations have not been explored in prior literature and go far beyond mere engineering integration.

---

> ### Author Response · Authors · 2025-11-20
> **General Response Part 3**
>
> ## **Q3: Lack of efficiency analysis (Reviewers $\color[RGB]{140,28,19}{\mathrm{dPaj}}$, $\color[RGB]{140,28,19}{\mathrm{hDE6}}$, $\color[RGB]{140,28,19}{\mathrm{Cg66}}$, and $\color[RGB]{140,28,19}{\mathrm{QuDJ}}$)**
>
> We appreciate the reviewer's attention to efficiency analysis. We fully agree that providing a systematic efficiency evaluation is crucial in the training-free setting. Accordingly, we have added detailed experimental results in Appendix A.6 of the revised PDF, as shown in the table below:
>
> | Methods       | Time(ms/image) ↓ | Memory(MB) ↓ | Performance(mIoU) ↑ |
> |---------------|------------------|--------------|---------------------|
> | Trident       | 89               | 2514         | 36.3                |
> | CorrCLIP      | 97               | 2890         | 33.5                |
> | SegEarth-OV   | 12               | 1392         | 39.1                |
> | **AlignCLIP-D** |                 |             |                    |
> | +SGA          | 12               | 1392         | 40.1                |
> | +CCE          | 16               | 2661         | 39.4                |
> | Ours          | 16               | 2661         | 41.3                |
> | **AlignCLIP-S**  |                 |             |                    |
> | +SGA          | 12               | 1392         | 40.1                |
> | +CCE          | 18               | 2782         | 39.3                |
> | Ours          | 18               | 2782         | 41.2                |
>
> Specifically, we evaluated the inference time and memory usage of two model variants on a single image using 8 × RTX 3090 GPUs. The results show:
> 1. **Inference speed**: The additional overhead compared to the SegEarth-OV is minimal (**AlignCLIP-D: +4ms**, **AlignCLIP-S: +6ms**), which is almost negligible.
> 2. **Memory usage**: Introducing the SGA and CCE modules slightly increases memory consumption, but achieves up to **+2.2% mIoU** performance improvement across multiple datasets, representing a reasonable and beneficial efficiency–performance trade-off.
>
> We hope that our response has adequately addressed your concerns. If you have any further questions or require additional clarification, please feel free to let us know.
>
> Thank you again for your time and valuable feedback.

---

### Author Response · Authors · 2025-12-01
**General Response to Area Chair**

Dear AC and Reviewers,

We deeply regret that the recent information-leak incident has affected the ICLR 2026 review process, and we fully understand the additional workload it has placed on the AC. Under such exceptional circumstances, we sincerely appreciate the time and extra effort the AC have devoted.

To help the AC more clearly understand our work and the responses we provided during the rebuttal phase, we first present a comprehensive summary of the core contributions of our work, followed by detailed replies to the major concerns raised by all reviewers.

---

Our work targets the task of **training-free open-vocabulary semantic segmentation (OVSS)** in remote sensing scenarios, where arbitrary categories can be segmented using only text prompts—without external data or additional training.

Our core contributions are summarized as follows:

- We are the **first to systematically reveal and quantitatively validate the key property that remote sensing images exhibit more compact intra-class feature distributions**, which serves as the foundational motivation for our method. We provide cross-modality quantitative analyses and visualizations to demonstrate the generality of this finding.
- We propose the SGA module, which establishes a training-free visual-to-text alignment mechanism based on visual prototypes. **This direction has been largely unexplored in prior training-free OVSS research** and **fills an important gap** by enabling optimization on the text modality.
- We propose the CCE module, which introduces a category-consistency constraint to **address inherent limitations of prior propagation-based methods**. This effectively prevents cross-category contamination and enhances the spatial consistency of the aligned features.
- Across eight remote sensing datasets, our model achieves up to **+2.2% mIoU** improvement on average. Moreover, SGA as a plug-in module that consistently enhances multiple training-free OVSS frameworks by **+1.6%~+1.9% mIoU**, demonstrating the independent value and practical applicability of our method.

---

**Responses to All Reviewers**

We sincerely thank all reviewers for their thoughtful comments and constructive feedback, which have been invaluable for improving our work. During the rebuttal phase, we provided detailed responses to every reviewer’s concerns. However, due to the impact of the leakage incident, we ultimately received a follow-up response from only one reviewer (**hDE6**). This reviewer **kept his positive score** and explicitly stated that **our responses addressed most of his concerns**. In addition, we further clarified the core innovations of our method to alleviate the reviewer’s questions regarding novelty, and we believe these clarifications sufficiently resolve such concerns.

We summarize the major issues raised by the reviewers during the rebuttal phase and the key improvements we made in response.

**1. Motivation and Quantitative Analysis**

We provide a more comprehensive and rigorous elaboration of our core motivation, namely that intra-class features in remote sensing images are more compact, supplementing evidence both qualitatively and quantitatively (line 85–99). Specifically, this includes:

- Quantitative comparison of cross-domain intra-class similarity (Table 1)
- Visualization of multi-class feature distributions (Figure 5)

We believe these experiments convincingly reinforce the reliability of our motivation.

**2. Methodological Novelty**

We further clarify the contributions of **AlignCLIP**, emphasizing that its design is **specifically motivated by the intra-class compactness of remote sensing images**, rather than a mere integration of existing modules:

- SGA is fully grounded in this motivation, aligning text features with visual prototypes and addressing a long-overlooked issue in OVSS from a cross-modal perspective. This approach fundamentally differs from existing methods that merely modify the CLIP visual encoder.
- Moreover, SGA can serve as a plug-in to enhance other OVSS frameworks (Table 10), demonstrating both the generality and independent value of the method.

**3. Efficiency Analysis**

We supplement our study with a systematic evaluation of inference time and memory consumption (Table 11), showing that:

- The inference time overhead increases by only **4~6 ms**, which is nearly negligible.
- Memory usage rises slightly but remains well within the acceptable range for modern GPUs.
- The method achieves up to **+2.2% mIoU** improvement, representing a clear positive trade-off between performance and efficiency.

We believe that our detailed replies have comprehensively addressed the reviewers’ concerns, and all corresponding revisions have been incorporated into the manuscript (**highlighted in red**). These improvements have significantly enhanced the quality of our work. We respectfully request that the AC take these thorough improvements into account when review our work.

Sincerely,

The Authors

---

### Note · Program_Chairs · 2026-01-17
**Submission Desk Rejected by Program Chairs**

The following references in this submission do not refer to real documents and/or have major errors in bibliographic information:

 X. Pan, Y. Li, J. Chen, and Z. Wang. Vdd: A new benchmark dataset for semantic segmentation of uav imagery. Remote Sensing, 13(7):1302, 2021. doi: 10.3390/rs13071302.